# Comprehensive multi-omics analysis reveals WEE1 as a synergistic lethal target with hyperthermia through CDK1 super-activation

Xiaohang Yang[1,2,3,8], Xingyuan Hu[1,2,8], Jingjing Yin[1,2,8], Wenting Li[1,2,4,8], Yu Fu[1,2], Bin Yang [1,2], Junpeng Fan [1,2], Funian Lu[1,2], Tianyu Qin [1,2], Xiaoyan Kang[1,2], Xucui Zhuang[1,2], Fuxia Li[4], Rourou Xiao[5], Tingyan Shi[6], Kun Song [3], Jing Li[7] ✉, Gang Chen [1,2] ✉ & Chaoyang Sun [1,2] ✉

Hyperthermic intraperitoneal chemotherapy's role in ovarian cancer remains controversial, hindered by limited understanding of hyperthermia-induced tumor cellular changes. This limits developing potent combinatory strategies anchored in hyperthermic intraperitoneal therapy (HIPET). Here, we perform a comprehensive multi-omics study on ovarian cancer cells under hyperthermia, unveiling a distinct molecular panorama, primarily characterized by rapid protein phosphorylation changes. Based on the phospho-signature, we pinpoint CDK1 kinase is hyperactivated during hyperthermia, influencing the global signaling landscape. We observe dynamic, reversible CDK1 activity, causing replication arrest and early mitotic entry post-hyperthermia. Subsequent drug screening shows WEE1 inhibition synergistically destroys cancer cells with hyperthermia. An in-house developed miniaturized device confirms hyperthermia and WEE1 inhibitor combination significantly reduces tumors in vivo. These findings offer additional insights into HIPET, detailing molecular mechanisms of hyperthermia and identifying precise drug combinations for targeted treatment. This research propels the concept of precise hyperthermic intraperitoneal therapy, highlighting its potential against ovarian cancer.

Epithelial ovarian cancer (EOC) generally presents at an advanced stage and remains the most lethal gynecological cancer[1]. Annually worldwide, 230,000 women will be diagnosed, and 150,000 will die from EOC[2]. One of the main factors contributing to the high death-to-incidence rate in EOC is the advanced stage at the time of initial diagnosis with widespread peritoneal metastasis (PM) and a lack of effective therapy for advanced disease[3]. Initial therapy for advanced EOC requires expert multidisciplinary care that includes surgery and adjuvant therapy[1,4]. The mainstay of therapy is "optimal surgery" that leaves minimal residual disease in combination with platinum- and

taxane-based chemotherapy[3,4]. While most women with advanced disease enter remission, most patients will experience disease recurrence and ultimately develop resistance to standard of care chemotherapy and succumb to their disease[1,4].

Hyperthermic intraperitoneal chemotherapy (HIPEC), which combines hyperthermia (HT) and intraperitoneal chemotherapy has been shown in a well-designed Dutch randomized trial (OVIHIPEC 1) to lead to a significant and substantial improvement in OS (45.7 vs. 33.9 months) in EOC patients who undergo optimal interval debulking surgery[5]. However, contradictory, or inconsistent levels of benefit

from HIPEC have been reported in other randomized controlled trials[6]. Moreover, the mechanism of the "HT" component in HIPEC is still poorly understood. Understanding how HT exerts beneficial effects will potentially shed light on acquired vulnerabilities[7,8] that could optimize combinatory strategies anchored in hyperthermic intraperitoneal therapy (HIPET).

Global transcriptional studies have provided important biological insights into HT-induced molecular pathways and signaling networks[9,10]. However, transcript levels do not always faithfully reflect protein abundance, especially during cell state transitions in response to environmental changes partly due to temporal delays between transcription and translation[11,12]. Moreover, post-translational regulation, especially phosphorylation, which plays a crucial role in regulating protein function and signaling for the fast adaptation to cellular stress, is poorly predicted from RNA levels[13,14]. Yet, systematic, and unbiased analyses of the signaling routes and subsequent functional consequences initiated by HT are still lacking.

Here, we show a global analysis of transcriptional, TMT-based whole proteome, and phosphoproteome changes induced by HT in ovarian cancer cells. Multi-tier integrative analyses illustrate the multiomics panorama of HT-induced molecular and signaling reprogramming. Most of the protein changes induced by HT are not reflected at mRNA levels, suggesting widespread post-transcriptional manipulation of proteins by HT, which validates the value of proteomic profiling. As predicted, the early molecular events during HT are better reflected by phosphoproteomics. Based on the phospho-signature, we identify Cyclin-Dependent Kinase 1 (CDK1) to be hyperactivated and responsible for the observed global signaling landscape during HT. Molecular and functional experiments demonstrate dynamic and reversible changes in CDK1 activity, subsequent replication arrest, and mitotic catastrophe after HT. Moreover, through data-driven drug screening, we discover synthetic lethal effects and further establish the optimal combination strategy and therapeutic windows of the combination of HT and WEE1 inhibition. Through an in-house developed miniaturized, streamlined, and reproducible device mimicking the clinical treatment protocol, we validate the in vivo efficacy of HIPET with WEE1 inhibitor (WEE1i) on murine ovarian cancer PM, indicating that combining WEE1i may afford an attractive therapeutic strategy for HIPET.

## Results

### Comprehensive multi-omics integrated analyses revealed hyperthermia resulted in CDK1 hyperactivation in epithelial ovarian cancer cells

Transcriptomics studies have been widely used to define signaling events driven by HT[15–17]. However, proteins are the key molecules that perform vital cellular functions[18]. Moreover, phosphorylation regulates protein activity, influences protein structure, and affects subcellular distribution[19]. Additionally, kinase/phosphatase signaling can manifest more rapidly than transcriptional changes, thus providing one of the earliest signatures in response to stimuli[20,21]. To obtain an unbiased perspective of HT-regulated signaling networks, we performed transcriptional sequencing coupled with proteomics and phosphoproteomics by multiplexed TMT and liquid chromatograph mass spectrometer (LC-MS)/MS approaches to quantify the transcriptome, proteome, and phosphoproteome in OVCAR8 cells (Fig. 1a), a cell line characterized by the common TP53 mutation in ovarian cancer[22–24], under both normothermic (37 °C) and hyperthermic (42 °C for 90 min) conditions, which are consistent with widely accepted parameters in previous studies and clinical research[25–27]. After filtering low-quality data, we quantified 16,954 transcripts, 6625 proteins, and 10,725 phosphorylation sites (Fig. S1a). Principal component analysis of the transcriptome, quantitative proteome, and phosphoproteome respectively showed high consistency between replicates (Fig. S1b).

Individually, differential expression (DE) analysis showed 692 transcripts (4.1 %), 322 proteins (4.9 %), and 2178 (20.3 %) phosphorylation sites as HT-responsive (Fig. S1c). Gene Set Enrichment Analysis (GSEA) showed that the HT-responsive transcripts converged on pathways related to response to heat or protein folding (Fig. S1d), which agreed with the previously described heat shock transcriptional response[28]. 6570 transcripts had also been identified in our proteomic profiling (Fig. S1e), demonstrating a significant overlap of genes/proteins discovered by the two profiles. Whereas the overall correlation of RNA and protein abundance was high, a poor correlation was uncovered between HT-induced transcriptome and proteome changes by directly comparing differential protein expression with mRNA changes at the same condition (Fig. S1f, g), consistent with widespread post-transcriptional regulation. Overall, compared with mRNA levels, the number of significantly regulated proteins upon HT was much less with 53 proteins increasing and 269 proteins decreasing (Fig. S1f), and only 14 differentially expressed RNAs and proteins (ten upregulated including HSPA6, HSPA1B, DEDD2, IER5, DUSP1, and four downregulated including DKK1 and CCN1) overlapped with the transcriptional response (Fig. S1f, g). Notably, of the few proteins that significantly increased in abundance, the vast majority were enriched in cell cycle pathways (E2F, G2M checkpoint, and Mitotic Spindle) (Fig. S1h, i). Considering that phosphorylation changes occur rapidly but protein changes require more time to develop, it is not surprising that the majority of significantly regulated molecular events occurred at the level of phosphorylation (Fig. S1c) as opposed to protein abundance (Fig. S1c). A total of 2178 phosphopeptides, corresponding to 1344 proteins, were differentially regulated in HT-treated cells, indicating that HT has a substantial impact on protein phosphorylation. Importantly, very few phosphosites (2.0 %) exhibited any change at the protein level (Fig. S1j), further suggesting that phosphorylation signaling represents a primary cell response over this time course of HT.

As our main goal was to reveal the unexplored synergistic lethal targets with HT, we next investigated the differential regulation of kinase signaling cascades and their effect on protein phosphorylation based on the well-known kinase and substrates relationship[29,30], which has been previously shown to perform well at estimating kinase activities[31,32]. The kinases predicted to be most strongly activated upon HT include several cell cycle kinases (CDK1, AURKB, and CDK2) (Fig. 1b). Kinases predicted to be downregulated include CK2A1, GSK3B, and MTOR (Fig. 1b). As shown in Fig. 1c, substrates for CDK1 (TOP1$^{pS394}$, TP53BP1$^{pT1609}$, CDC25C$^{pT48}$, USP14$^{pT235}$, etc) were significantly overrepresented in the upregulated phosphoproteins after HT.

### Hyperthermia results in CDK1-dependent cell cycle perturbation, replication stress, and DNA damage

Integrative analysis showed the CDK1-dependent cell cycle checkpoint was the most perturbed pathway after HT, including its own inhibitory Tyr15 (Y15) phosphorylation were significantly decreased (Fig. 1d). To validate HT-induced CDK1 hyperactivation and resulting molecular and functional perturbations, we confirmed that HT induces time-dependent decreases of the inhibitory phosphorylation of CDK1 on Thr14 (T14) and Y15 without marked effects on total protein levels in OVCAR8, A2780, and ID8 cells (Fig. 1e). Phospho-Thr320 PP1Cα (PP1Cα-pT320), which has been proposed as an indicator of CDK1 activity[33], increased gradually in a time-dependent manner after HT as assessed by western blotting (Fig. 1e). In addition, phospho-Ser139 H2AX (γH2AX) was massively increased indicating HT-induced replication stress eventually led to DNA damage (Fig. 1e).

CDK1 controls critical cell cycle events, such as DNA replication and G2/M-phase transition[34,35]. Excessive activation of CDK1 induces replication stress through CDK1-dependent aberrant firing of replication origins[36]. Thus, we tracked the process of cell cycle progression and DNA replication after HT by 5-Ethynyl-2′-deoxyuridine (EdU) pulse

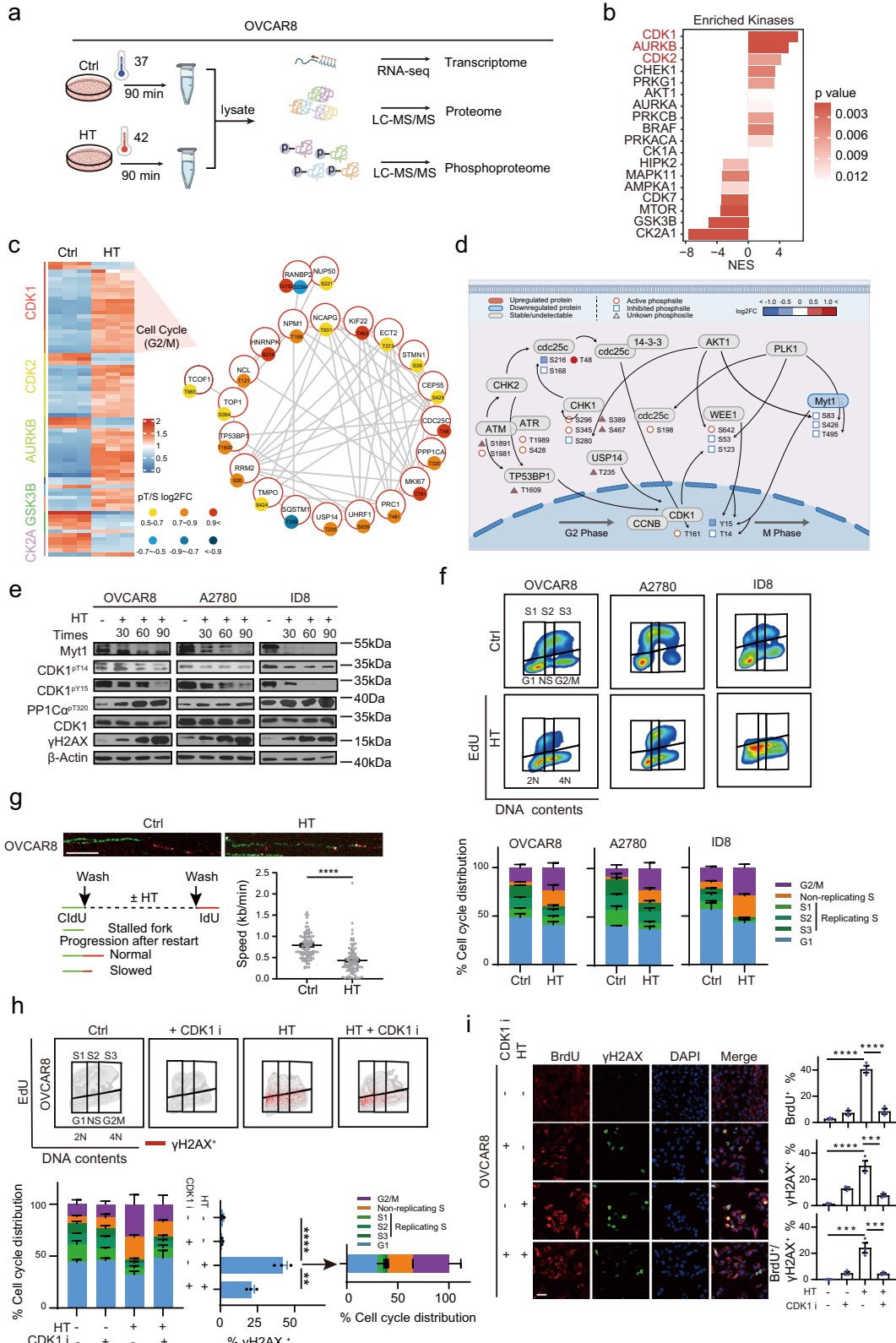

labeling flow cytometry. Consistent with the critical role of CDK1 on the G2/M cell cycle checkpoint, HT increased G2/M cell populations (Fig. 1f). In addition, we observed that HT forced OVCAR8, A2780, and ID8 cells to accumulate non-replicating S-phase cells (NS phase, exhibiting a DNA content between 2N and 4N but not incorporating the synthetic nucleoside EdU), evidence of DNA replication stress occurring after HT (Fig. 1f). Second, we depicted the progression of

individual replication forks by DNA fiber assay, the gold standard assay to monitor fork speed progression and origin firing. Fork velocity was significantly reduced upon HT treatment (from an average of 0.6–0.8 kb/min to 0.3–0.5 kb/min) in OVCAR8 cells (Fig. 1g), further supporting HT induces replication fork stalling and collapse. Stalled DNA replication forks lead to the exposure of single-stranded DNA (ssDNA)[37,38].

**Fig. 1 | Hyperthermia resulted in CDK1-dependent cell cycle perturbation, replication stress, and DNA damage. a** Experimental scheme of the transcriptomic, proteomic, and phosphoproteomic profiling. LC-MS/MS: Liquid Chromatography-Tandem Mass Spectrometry ($n = 3$ per group). **b** Bar plot showing differential phosphosites when comparing hyperthermia (42 °C, HT) to normothermia (37 °C, Ctrl) treated OVCAR8 cells, as analyzed by PTM-SEA. **c** Heat map showcasing the differentially regulated phosphorylation sites associated with kinases enriched in part (**b**) (left). Protein-protein interaction network of differential substrates of CDK1, as sourced from the STRING database (right). **d** Overview of the changes in protein and phosphorylation levels associated with CDK1-dependent cell cycle induction by HT. **e** OVCAR8, A2780, and ID8 cells were treated with HT for 30, 60, and 90 min. A representative western blot image is shown with indicated proteins. **f** Cell cycle distribution after exposure to HT for 90 min in OVCAR8, A2780, and ID8 cell lines ($n = 3$ per group). **g** DNA fiber analysis of OVCAR8 cells treated as in part (**f**) yielded representative replication fork images with a scale bar of 10 μm. Fork velocity distribution analysis was based on 100 ongoing replication forks pooled from three independent experiments. **h** Cells were pretreated with or without a CDK1 inhibitor (CDK1i, RO-3306, 5 μM) for 48 h prior to exposure to HT for 90 min. γH2AX, EdU, and PI staining were analyzed by flow cytometry in OVCAR8 cells ($n = 3$ per group). The quantification of γH2AX positive cells in various cell cycle phases is shown. **i** To detect ssDNA, cells were cultured in a medium with 10 μM BrdU throughout the experiment; DNA was not denatured before staining against BrdU. Representative images and quantification of OVCAR8 cells stained with BrdU, γH2AX, and DAPI are shown ($n = 3$ per group). Scale bar, 25 μm. Comparisons were performed by unpaired two-tailed Student's $t$ test in (**g**), one-way ANOVA followed by Dunnett's multiple comparisons test in (**h**), and one-way ANOVA followed by Tukey's multiple comparisons test in (**i**). Data are presented as mean ± SEM. ****$p < 0.0001$, ***$p < 0.001$, **$p < 0.01$. All analytical data are derived from a minimum of three biologically independent experiments, with "$n$" indicating the specific number of replicates. Source data are provided as a Source Data file.

To further validate that hyperactivated CDK1 is responsible for the cycle perturbation, replication stress, and DNA damage accumulation after HT, a CDK1 inhibitor (CDK1i, RO-3306) was added to cells treated with HT. RO-3306 restored cell viability in multiple ovarian cancer cell lines after HT therapy (Fig. S1k). In addition, RO3306 abolished HT-induced G2/M cell population accumulation and alleviated non-replicating S-phase, and replication stress (Fig. 1h and Fig. S1l). Notably, HT-induced DNA damage (γH2AX) was mainly distributed in G2/M and nonreplication S phase (Figs. 1h, S1l), consistent with replication arrest and mitotic catastrophe leading to DNA damage. Moreover, a massive accumulation of ssDNA was observed after HT in OVCAR8 cells with BrdU staining under non-denaturing conditions (Fig. 1i). Notably, numerous BrdU positive cells displayed concurrent diffuse pan-nuclear γH2AX staining instead of punctate foci (Fig. 1i), indicating catastrophic replication stress and irreversible cell death[39]. As expected, RO-3306 significantly blocked γH2AX and ssDNA induction (Fig. 1i). Together, these data suggest that HT enhanced DNA replication stress and abrogate the G2/M cell cycle checkpoint in a CDK1-dependent manner, indicating a potential synergistic effect with WEE1 inhibitors which might further augment CDK1 activity[40].

To determine generalizability, we detected molecular changes after HT in an additional 13 cell lines with various genetic backgrounds (eight ovarian cancer, two colon cancer, and three non-tumor cells) (Fig. S1m). Strikingly, phospho-Tyr15 CDK1 (CDK1-pY15) was downregulated, and DNA damage was increased upon HT in all cancer cells irrespective of genotype and cell lineage. In contrast, no significant changes of CDK1-pY15 and γH2AX were observed in non-tumor IOSE80 (normal ovarian surface epithelial), HUVEC (human umbilical vein endothelial), and HaCaT (human keratinocyte) cells following HT treatment (Fig. S1m). Given that heat shock transcription factor 1 (HSF1) mediates the transcription of heat shock proteins integral to cellular stress responses[41], we utilized CRISPR-Cas9 to establish an HSF1 knock-out SKOV3 cell (Fig. S1n). Our findings demonstrate that the CDK1 activation after HT was independent of HSF1 (Fig. S1o).

### Hyperthermia promotes dNTPs depletion which is responsible for replication stress and DNA damage accumulation

dNTP homeostasis plays an essential role in cell fate following CDK1-induced replication stress with dNTPs starvation ultimately leading to replication collapse and cell death[42]. RRM2, a ribonucleotide reductase subunit, is critical for maintaining dNTP homeostasis preventing replication catastrophe in response to replication stress[43,44]. CDK1 phosphorylates RRM2 at Thr33 and promotes RRM2 ubiquitylation and degradation via the E3 ubiquitin ligase SCF (Cyclin F)[45]. Therefore, we examined dynamic RRM2 expression and found that RRM2 displayed a gradual decrease after HT in OVCAR8, A2780, and ID8 cells (Fig. 2a) which was synchronous with the decrease in CDK1 phosphorylation on T14 and Y15 after HT (Fig. 1e). To confirm that extensive replication stress, DNA damage accumulation after HT are on-target responses to nucleotide pool depletion by CDK1 hyperactivation, we reasoned dNTPs (dNs) supplementation would rescue HT-induced replication stress and DNA damage. As indicated, replenishing dNTP pools alleviated HT-induced cell death in multiple cancer cell lines (Fig. 2b). Functionally, exogenous dNTPs were sufficient to diminish non-replicating S and G2/M cells induced by HT (Fig. 2c). Furthermore, HT-induced γH2AX positive cells, predominantly at non-replicating S and G2/M phase, were alleviated by dNTPs supplementation (Fig. 2c). ssDNA and pan-nuclear γH2AX staining further validated dNTPs supply dramatically blunted HT-induced replication stress and DNA damage in OVCAR8 cells (Fig. 2d). These results collectively suggest that dNTPs depletion contributes to the DNA replication stress and DNA damage accumulation after HT.

### The dynamic changes of CDK1 activation and recovery after hyperthermia

Diverse molecular events or biological pathways induced by HT exhibit dynamic changes[46,47]. Thus, uncovering the dynamics of CDK1 activity after HT will not only help us to understand HT-induced replication stress and DNA damage but also provide crucial information about the "therapeutic window" of the druggable vulnerability conferred by HT. After treatment with HT at 42 °C for 90 min, CDK1 activities were strongly activated, gradually increased, and sustained for several hours (3, 6, and more than 24 h in OVCAR8, A2780, and ID8 cells respectively) after return to 37 °C (Fig. 3a). Notably, HT-induced CDK1 hyperactivation seems reversible although the recovery time varied in different cells (6, 12, and more than 48 h in OVCAR8, A2780, and ID8 cells respectively) (Fig. 3a). Impressively, DNA damage represented by γH2AX, and cell apoptosis indicated by cleaved caspase-3 were synchronously increased and recovered with the change of CDK1 activity, strongly indicating a causal relationship between them (Fig. 3a). In ID8 cells, HT-induced CDK1 activation, DNA damage, and apoptosis were sustained for up to 48 h, possibly because these cells exhibit a higher sensitivity to HT (Fig. 3a).

Functionally, EdU and γH2AX co-labeling flow cytometry (Fig. 3b) revealed the dynamic replication stress and subsequent DNA damage kinetics after HT. As expected, the non-replicating S and G2/M phase cells increased and were sustained for 3 and 6 h respectively in OVCAR8 and A2780 cells, and recovered at 6 and 12 h respectively (Figs. 3b, S2a). Again, the γH2AX positive cells, which were predominantly non-replicating S (indicating cells were undergoing replication catastrophe) and G2/M phase (a marker of premature entry into mitosis while cells are still undergoing DNA damage) gradually increased, peaked at 3 h, and recovered at 6 h after HT in OVCAR8 cells (Fig. 3b, c). Immunofluorescence co-staining of ssDNA and γH2AX (Fig. 3d) confirmed the presence of ssDNA positive and ssDNA-γH2AX double-positive cells which increased and were maintained for

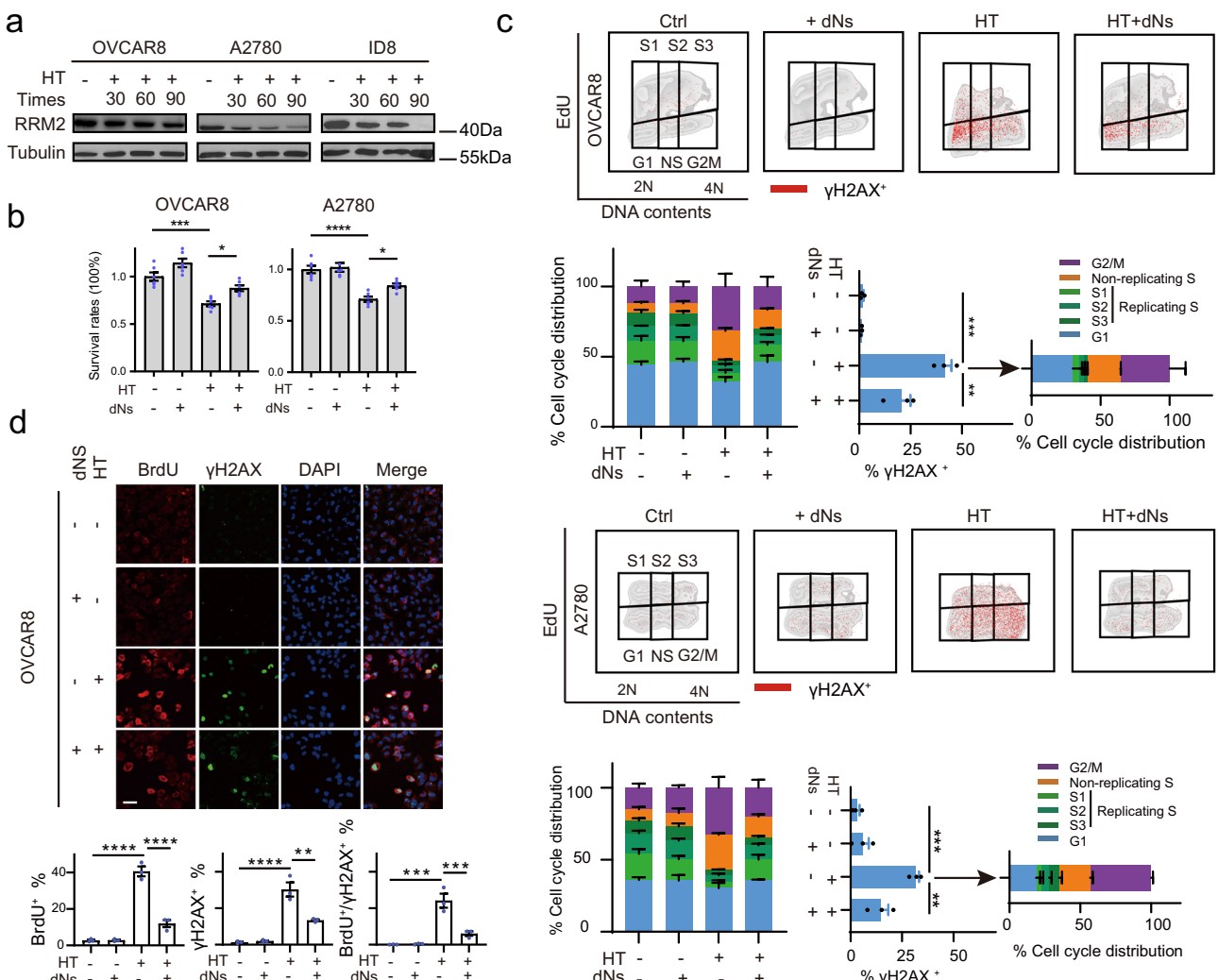

**Fig. 2 | Hyperthermia promotes dNTPs depletion which is responsible for replication stress and DNA damage accumulation. a** OVCAR8, A2780, and ID8 cells were treated with hyperthermia (HT, 42 °C) for 30, 60, and 90 min. Representative western blot image illustrates dynamic RRM2 levels at different heating durations. **b** Cells were cultured with or without dNTPs mix (dNs, 50 μM) for 48 h prior to being treated with HT for 90 min. Changes in cell viability were determined by the CCK8 assay (*n* = 6 per group). **c** γH2AX, EdU, and PI staining were analyzed by flow cytometry in OVCAR8 and A2780 cells after exposure to HT with or without dNs (*n* = 3 per group). Quantification of γH2AX positive cells in various cell cycle phases is presented. **d** Representative images and quantification of BrdU and γH2AX positive OVCAR8 cells after being treated as in part b are presented (*n* = 3 per group). Scale bar, 25 μm. Comparisons were performed by one-way ANOVA followed by Dunnett's multiple comparisons test in (**c**) and one-way ANOVA followed by Tukey's multiple comparisons test in (**b**, **d**). Data are presented as mean ± SEM. ****$p < 0.0001$, ***$p < 0.001$, **$p < 0.01$, *$p < 0.05$. All analytical data are derived from a minimum of three biologically independent experiments, with "*n*" indicating the specific number of replicates. Source data are provided as a Source Data file.

3, 6, and more than 24 h in OVCAR8, A2780, and ID8 cells respectively (Figs. 3d, S2b). Cell viability by cell counting kit 8 (CCK8) and apoptosis detected by Annexin V-FITC/PI assays further confirm dynamic changes in cell survival and apoptosis that were synchronous with the dynamic changes of CDK1 activation after HT (Fig. S2c, d).

Collectively, these results uncovered HT-induced dynamic and reversible changes of CDK1 activation, replication stress, DNA damage, and ultimately cell death after cells returned to normal temperature. These reversible dynamic molecular changes indicate a constraint on the therapeutic efficacy of transient mono-thermal therapy in ovarian cancer.

**Hyperthermia sensitizes to WEE1 inhibitor by augmenting replication stress, DNA damage, and inducing mitotic catastrophe**

The "one-two punch" approach in cancer therapy entails sequentially applying two distinct treatments or interventions to augment the therapeutic outcome. The initial "punch" constitutes a primary treatment targeting tumor cells, whereas the subsequent "punch" serves as a secondary intervention capitalizing on induced vulnerabilities or alterations caused by the initial treatment. Under the conceptual model of the "one-two punch" approach, we further explore whether this HT-induced molecular characteristic would generate a druggable addiction manipulable by a secondary perturbation. As aforementioned, HT-induced CDK1 hyperactivation, G2/M phase perturbation, replication stress, and DNA damage (Fig. 1). Thus, we performed candidate drug screening of synergistic potential through a combination of HT with 15 well-characterized inhibitors targeting or capitalizing on CDK1 related G2/M cell cycle checkpoint, replication stress or DNA damage pathways individually in OVCAR8, and A2780 cells (Fig. 4a, b). Meanwhile, the combination of (DDP) and paclitaxel (TAX), which is a standard component of HIPEC, was used as the positive control.

Here, we applied an emerging method to evaluate the sensitization effects of HT on individual inhibitors: the ΔAUC (the area under the fitted dose-response curve)% = AUC (Ctrl) − AUC (HT)/AUC (Ctrl)

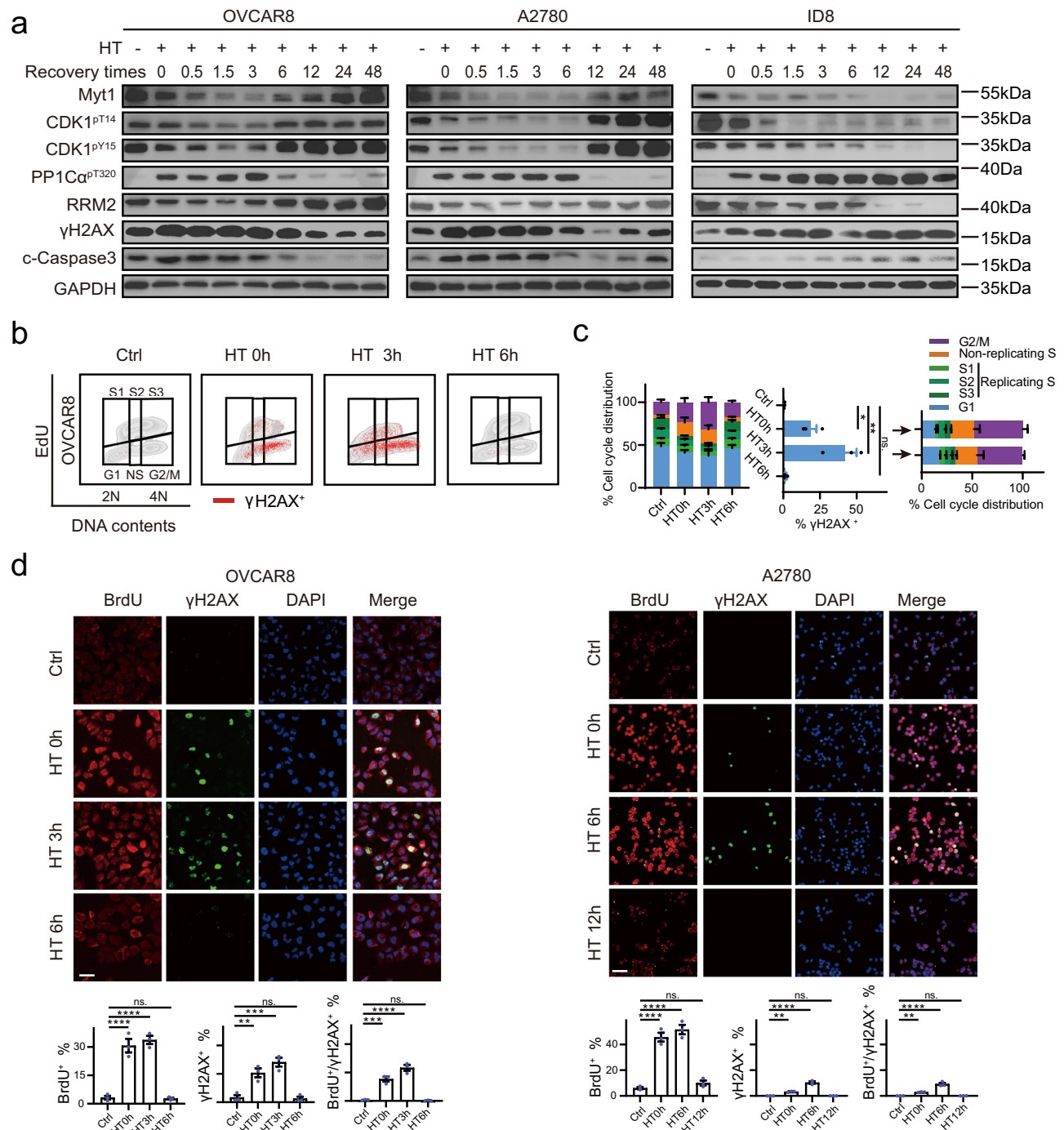

**Fig. 3 | The dynamic changes of CDK1 activation and recovery after hyperthermia. a** Representative western blot image with indicated proteins, delineating the dynamic changes in CDK1 activity at different recovery times post-hyperthermia (HT). OVCAR8, A2780, and ID8 cells were treated with HT (42 °C) incubation for 90 min, then were allowed to recover at 37 °C for the indicated times (0, 0.5, 1.5, 3, 6, 12, 24, and 48 h). **b, c** γH2AX, EdU, and PI staining were analyzed by flow cytometry in OVCAR8 cells that were treated with HT for 90 min and allowed to recover at 37 °C for the indicated times (n = 3 per group). **d** Representative images and quantification of BrdU and γH2AX positive OVCAR8 and A2780 cells

treated with HT for 90 min, followed by recovery at 37 °C for the indicated times, are presented (n = 3 per group). Scale bar, 25 μm. Comparisons were performed by unpaired two-tailed Student's t test in (**c**) and one-way ANOVA followed by Dunnett's multiple comparisons test in (**d**). Data are presented as mean ± SEM. ****p < 0.0001, ***p < 0.001, **p < 0.01, *p < 0.05, ns., not significant. All analytical data are derived from a minimum of three biologically independent experiments, with "n" indicating the specific number of replicates. Source data are provided as a Source Data file.

(Fig. 4c), and higher ΔAUC% implies greater synergistic potential between HT and drugs. The combination of HT with WEE1i (AZD1775) displayed the most consistent and strongest synergism both in OVCAR8 and A2780 cells, which exceeded the DDP and TAX combination (Fig. 4d). Notably, to avoid overshadowing the effect of HT with excessive standalone impact of the drug, we carefully selected the

minimal concentration of AZD1775 required to activate CDK1 in each cell line for subsequent experiments, as determined by our previous research[48,49]. Colony formation assay confirmed the synergistic activity of concurrent HT and WEE1i in OVCAR8, A2780, and ID8 cells (Fig. S3a). Furthermore, the combination treatment led to a pronounced increase in cell apoptosis detected by Annexin V-FITC/PI assay when compared

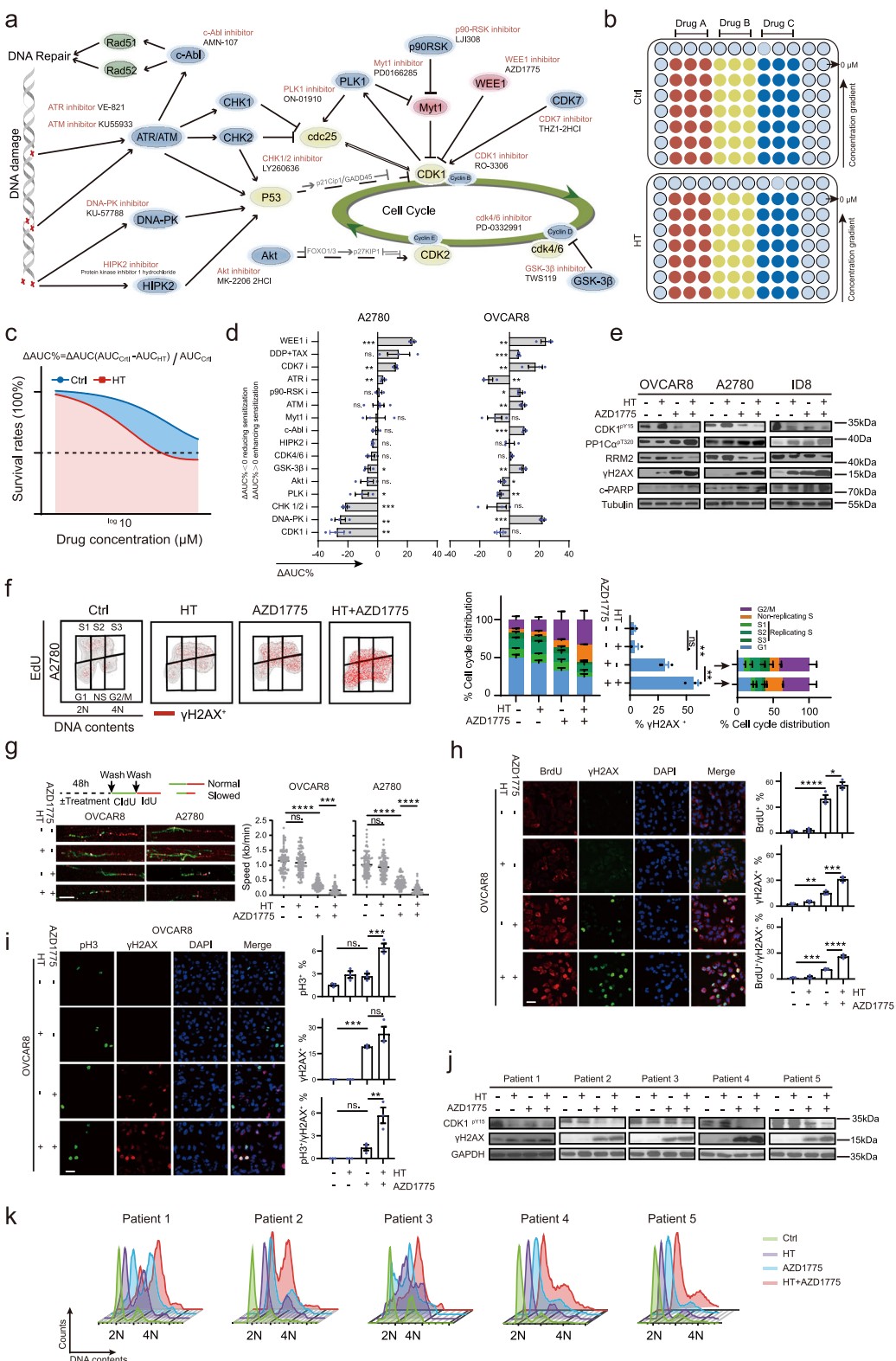

to WEE1 inhibition alone, confirming the observed synergistic cyto-toxicity (Fig. S3b). Similar results were observed in an additional nine ovarian cancer and two colon cancer cell lines, supporting the gen-eralizability of the synergism between HT and WEE1 inhibition (Fig. S3c). Despite the high specificity of AZD1775[40,50], we further employed small interfering RNA (siRNA)-mediated WEE1 knockdown in OVCAR8 cells to confirm that the synergy between HT and WEE1

inhibition is not solely attributed to AZD1775's off-target actions (Fig. S3d, e).

High-grade serous ovarian carcinoma (HGSOC), characterized by ubiquitous TP53 mutations, exhibits G1/S checkpoint deficiencies[22], suggesting that abrogating the G2/M checkpoint with WEE1i might selectively target TP53-defective ovarian cancer cells[51]. Interestingly, as demonstrated in Fig. S3c, HT-induced CDK1 activation, and the

**Fig. 4 | Hyperthermia sensitized WEE1 inhibitor by augmenting replication stress, deteriorating DNA damage, and inducing mitotic catastrophe.**
**a** Functionalities and trade names of cell cycle-related inhibitors in synergistic drug screening with hyperthermia (HT). **b** A flow diagram of drug screening is presented. Cells seeded into 96-well plates were treated with increasing doses of inhibitors for 48 h in the presence or absence of HT (42 °C) for 90 min, and cell viability was estimated using the CCK8. **c** The sensitizing effect of HT on drugs is defined as ΔAUC% = AUC(Ctrl) − AUC(HT)/AUC(Ctrl). **d** The ΔAUC% for the combination of HT with various drugs was calculated ($n = 4$ per group in ATMi, $n = 3$ in others). **e** Representative western blot results for the combination of HT with AZD1775. OVCAR8, A2780, and ID8 cell lines were treated with AZD1775 (250, 800, and 800 nM, respectively) for 48 h, with or without a 90 min HT treatment during the initial drug exposure. **f** γH2AX, EdU, and PI staining were analyzed by flow cytometry in A2780 cells treated as in (**e**) ($n = 3$ per group). **g** OVCAR8 and A2780 cells, treated as in (**e**), underwent DNA fiber analysis with 100 fibers collected from three

independent experimental replicates for each group. Scale bar, 10 μm.
**h** Representative images and quantification of BrdU and γH2AX positive OVCAR8 cells treated as in (**e**) are shown ($n = 3$ per group). Scale bar, 25 μm. **i** Representative images and quantification of pH3 and γH2AX positive OVCAR8 cells treated as in **e** are shown ($n = 3$ per group). Scale bar, 25 μm. **j** Representative western blot results depict the combination of HT with AZD1775 in five primary HGSOC samples across five independent experiments. **k** Cell cycle flow cytometric analysis was performed on five primary HGSOC samples across five independent experiments, as described in (**j**). Comparisons were performed by unpaired two-tailed Student's *t* test in (**d**) and one-way ANOVA followed by Tukey's multiple comparisons test in (**f, g, h, i**). Data are presented as mean ± SEM. ****$p < 0.0001$, ***$p < 0.001$, **$p < 0.01$, *$p < 0.05$, ns., not significant. All analytical data are derived from a minimum of three biologically independent experiments, with "$n$" indicating the specific number of replicates. Source data are provided as a Source Data file.

observed synergism between HT and WEE1i were consistent across various cell lines, regardless of their TP53 status. Furthermore, TP53 knock-out using CRISPR-Cas9 in TP53-WT A2780 and ID8 cells confirmed that the functional status of TP53 does not influence the synergistic effect between HT and WEE1 inhibition (Fig. S3f, g). These findings suggest a potential broader clinical application for WEE1i when used in conjunction with HT.

Interestingly, WEE1 inhibits the activities of CDK1 by phosphorylating Y15[52]. WEE1i induces replication stress and cell death through CDK1-dependent aberrant firing of replication origins and unscheduled mitotic entry[37,53,54]. Therefore, we hypothesized that the synergism between HT and WEE1i might be reliant on their effect on CDK1 activity. Indeed, in line with the cell viability and apoptosis assays, the combination of HT and WEE1i augmented CDK1 activation, as indicated by a further reduction in CDK1-pY15 and an increase in PP1Cα-pT320 (Fig. 4e). Moreover, the combination of HT and WEE1i increased DNA damage (γH2AX), and apoptosis (cleaved PARP) as compared with WEE1i mono-therapy (Fig. 4e). Of note, it is worth mentioning that, like the results shown in Fig. 3, the induction of CDK1 activation, DNA damage, and apoptosis by HT monotherapy was restored in OVCAR8 and A2780 cells but persisted in ID8 cells for 48 h (Fig. 4e). After treatment for 48 h, we colabeled cells with EdU, PI, and γH2AX. As expected, while AZD1775 mono-therapy modestly increased nonreplicating S and G2/M phase cells, the combinational therapy further augmented these phenotypes, indicating DNA replication stress and G2/M checkpoint failure (Fig. 4f). The γH2AX positive cells, predominantly in the nonreplication S and G2/M phase, were further potentiated in the combination (Fig. 4f). Furthermore, while AZD1775 significantly decreased the speed of the DNA replication fork after 24 h of treatment, the combination of HT and AZD1775 nearly halted fork progression (average fork velocity <0.2 kb/min) (Fig. 4g). Similar results were observed in OVCAR8 cells with ssDNA and γH2AX co-staining as a further indication of replication stress and subsequent DNA damage (Figs. 4h, S3h). Furthermore, while WEE1i modestly increased the proportion of histone H3 Ser10 phosphorylation (pH3) mitotic cells, concurrent HT and WEE1 inhibition massively increased the proportion of pH3+ cells (Figs. 4i, S3i). Most importantly, concurrent HT and WEE1i treatment markedly increased the proportion of pH3 and γH2AX double-positive cells whereas WEE1i mono-therapy only slightly increased these double-positive cells (Figs. 4i, S3i). The accumulated entry into mitosis with persistent DNA damage after combinational therapy (Figs. 4i, S3i) would eventually lead to mitotic catastrophe and cell death[55–57].

To validate the clinical relevance of our findings, we further test the synergism of HT and AZD1775 in five patient-derived primary cell cultures from HGSOC, with patient clinical details presented in Supplementary Table 1. Consistent with the role of WEE1 as a critical mediator of the S phase and G2/M phase checkpoints, AZD1775 mono-therapy resulted in modest DNA damage (Fig. 4j) and M cell

accumulation (Fig. 4k). In all tested primary ovarian cancer cells, AZD1775 combined with HT further intensified these events (Fig. 4j, k), albeit with varying responses among different patients due to inter-patient variability. Together, results from the different approaches above strongly suggest a synthetic lethal interaction between HT and WEE1 inhibition.

## CDK1 hyperactivation and dNTPs starvation mediate the synergism between HT and WEE1 inhibitor
If the CDK1 hyperactivation and dNTPs depletion (as a result of CDK1 hyperactivation) were the cause of the synergism between HT and WEE1 inhibition, then CDK1 inhibition should rescue the synthetic lethality between AZD1775 and HT. Indeed, CDK1 inhibition with RO-3306 not only alleviated AZD1775-induced cell death but also abolished the synthetic lethality between AZD1775 and HT (Fig. 5a). Likewise, exogenous dNTPs supplementation also rescues, although to a lesser extent, the synthetic lethality of AZD1775 and HT treatment (Fig. 5a). Functionally, the inhibition of CDK1 alleviated the effects of WEE1i monotherapy and the combination therapy-induced DNA damage and replication catastrophe (Fig. 5b–d), underscoring the pivotal role played by CDK1 activation in the synergism between HT and WEE1i.

dNTP homeostasis is essential to prevent replication catastrophe in response to CDK1-induced replication stress[42,58]. Thus, increased CDK1 activation through HT and WEE1 inhibition might result in critically low dNTPs levels and consequent replication collapse and cell death. Indeed, dNTPs supply also abrogated the synthetic lethal interaction between HT and WEE1i (Fig. 5e–g). The data together suggested that CDK1 hyperactivation and consequential dNTPs depletion at least partly contribute to the vulnerability of HT-induced cells to WEE1 inhibition.

## Combined modality strategies and therapeutic windows for the combinational therapy of hyperthermia and WEE1 inhibition
In the "one-two punch" approach, sequencing and timing the interventions are crucial to achieve optimal synergistic anti-tumor effects, ensuring the second intervention is promptly administered following the maximum disruption from the first intervention. The above results have shown that the simultaneous application of HT in combination with WEE1i can produce synthetic lethal effects, but the reversible dynamic molecular changes induced by HT provide the limits of the "therapeutic window" for targeting the vulnerabilities induced by HT. Therefore, further investigations are necessary to identify the optimal combination strategy that incorporates these dynamic changes and maximizes the therapeutic window, thereby enhancing treatment efficacy.

Since we have identified that HT selectively enhances the anti-tumor response to WEE1 inhibition through replication arrest and mitotic catastrophe, we questioned which combined modality,

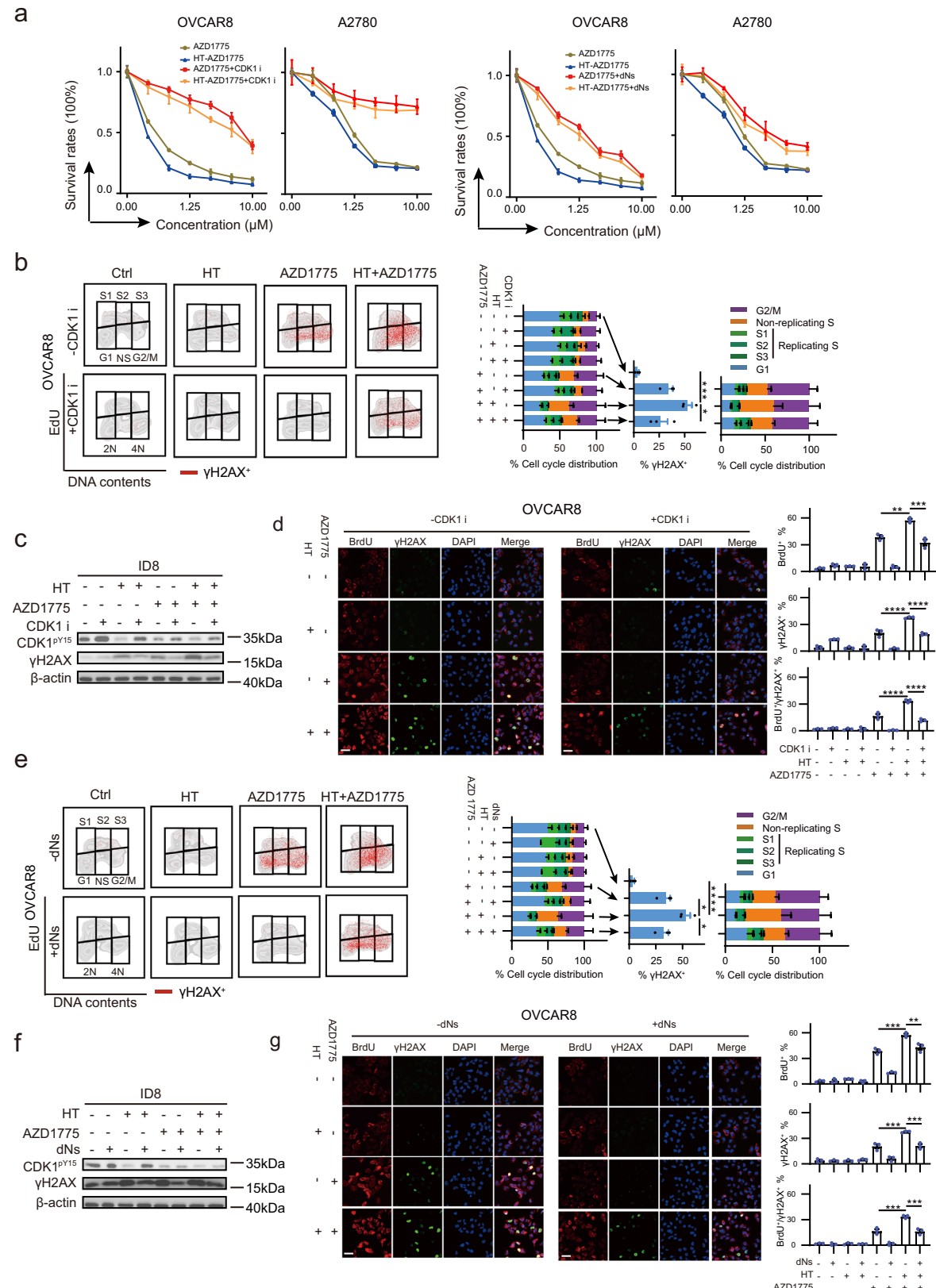

sequential or concurrent delivery of these therapies, will be the optimal combination strategy (Fig. 6a). Intriguingly, whether administered concurrently or sequentially, with HT preceding WEE1i, a pronounced synergistic effect was observed in killing OVCAR8, A2780, and ID8 cells (Fig. 6b, c). Conversely, when the order of administration was reversed, with WEE1i preceding HT, the synergistic effect was not apparent. We

next explored the therapeutic window for the combinational therapy of HT and WEE1 inhibition (Fig. 6d). Consistent with HT-induced dynamic CDK1 activation and recovery in various cells, the synergistic lethal effects of sequential combination were observed when HT was applied before WEE1 inhibition, with optimal therapeutic windows of within 6 h for OVCAR8 cells, within 12 h for A2780 cells, and extending

**Fig. 5 | CDK1 hyperactivation and dNTP starvation mediated the synergism between HT and WEE1 inhibitor. a** Dose–response curves for AZD1775 monotherapy or concurrent HT-AZD1775 therapy were generated in OVCAR8 and A2780 cell lines, treated with graded concentrations for 48 h, in the presence or absence of a CDK1 inhibitor (CDK1i, 5 μM) or dNTPs mix (dNs, 50 μM) ($n = 3$ per group). **b** OVCAR8 cells were treated with AZD1775 (250 nM) for 48 h in the presence or absence of hyperthermia (HT, 42 °C, 90 min), combined with or without CDK1i (5 μM). γH2AX, EdU, and PI staining results analyzed by flow cytometry are shown ($n = 3$ per group). **c** Representative western blot images in ID8 cells treated as in (**b**) (AZD1775, 800 nM). **d** Representative images and quantification of BrdU and γH2AX positive OVCAR8 cells treated as in (**b**) are shown ($n = 3$ per group). Scale bar, 25 μm. **e** OVCAR8 cells were treated with AZD1775 (250 nM) for 48 h in the presence or absence of HT (42 °C, 90 min), combined with or without dNs (50 μM). γH2AX, EdU, and PI staining results analyzed by flow cytometry are shown ($n = 3$ per group). **f** Representative western blot images in ID8 cells treated as in (**e**) (AZD1775, 800 nM). **g** Representative images and quantification of BrdU and γH2AX positive OVCAR8 cells treated as in (**e**) are shown ($n = 3$ per group). Scale bar, 25 μm. Comparisons were performed by one-way ANOVA followed by Dunnett's multiple comparisons test in (**b, e**) and one-way ANOVA followed by Tukey's multiple comparisons test in (**d, g**). Data are presented as mean ± SEM. ****$p < 0.0001$, ***$p < 0.001$, **$p < 0.01$, *$p < 0.05$. All analytical data are derived from a minimum of three biologically independent experiments, with "$n$" indicating the specific number of replicates. Source data are provided as a Source Data file.

to 48 h or even longer for ID8 cells (Fig. 6e, f). These findings highlight the importance of considering the timing and sequence of treatments to maximize the synergistic potential of combining HT with a WEE1i. Importantly, our results highlight the significance of initiating therapy with HT as the initial intervention, followed by WEE1 inhibition, to maximize the synergistic killing effect specifically in these cell lines.

Collectively, these data not only support the concept that HT-induced CDK1 activation created the vulnerability for WEE1 inhibition, but also provide the effective combined strategy and optimal "therapeutic window" for combination.

### Myt1 downregulation contributes to the hyperthermia-induced CDK1 hyperactivation and the synthetic lethality with WEE1 inhibition

To explore the upstream molecule responsible for the CDK1 activation after HT, we rechecked our proteomic data and noticed that Myt1, another inhibitory CDK1 kinase in addition to WEE1, was significantly downregulated after HT (fold change = 0.769 $p = 0.0008$). While WEE1 phosphorylates CDK1 on Y15 to inhibit CDK1 activation, Myt1, which is structurally related to WEE1, negatively regulates CDK1 by its inhibitory phosphorylation on T14/Y15 and sequestration of CDK1 in the cytoplasm[59]. Therefore, we reasoned that Myt1 downregulation might contribute to HT-induced CDK1 hyperactivation and contribute to the observed synthetic lethality with WEE1 inhibition. First, consistent with the gradual decrease of CDK1-pY15, Myt1, and CDK1-pT14 (another downstream target of Myt1) were also downregulated in a time-dependent fashion after HT (Fig. 1e). Second, paralleling the dynamic trend of CDK1-pY15, Myt1, and CDK1-pT14 were synchronously decreased and recovered after HT (Fig. 3a). More importantly, the dynamic changes of Myt1 and CDK1-pY15 were observed in five primary HGSOC cultures although the persistence and recovery time varies in different patients (Fig. 7a).

To corroborate that Myt1 downregulation was mainly responsible for HT-mediated CDK1 hyperactivation, we stably overexpressed Myt1 in OVCAR8 and ID8 cells which markedly increased inhibitory CDK1-pY15 and CDK1-pT14 phosphorylation (Fig. 7b). Furthermore, overexpression of Myt1 diminished HT-induced DNA damage (as indicated by reduced γH2AX accumulation) and cell death in ID8 and OVCAR8 cells (Fig. 7c, d). Functionally, Myt1 ectopic expression decreased the HT-induced non-replicating S phase and G2/M phase population (Fig. 7e) and diminished ssDNA and replication catastrophe after HT (Fig. 7f). Hence, Myt1 downregulation at least partly contributes to the HT-induced CDK1 hyperactivation, replication stress, and DNA damage. Notably, the downregulation of Myt1 induced by HT also occurs independently of HSF1 (Fig. S1o).

Previous studies showed a high level of Myt1 is associated with intrinsic AZD1775 resistance, and compensatory Myt1 activation promoted acquired resistance of cancer cells to WEE1 inhibition[60]. We verified that Myt1, CDK1-pY15, and CDK1-pT14 were significantly elevated in WEE1i-resistant cells (ID8R) which showed a 10-fold increase of IC50 when compared to its parental counterpart (ID8 cells) (Fig. 7g, h). Meanwhile, Myt1 downregulation by siRNA resensitized ID8R cells to AZD1775 (Fig. 7i, j). Moreover, ectopic Myt1 expression rendered parental ID8 and OVCAR8 cells resistant to AZD1775, which reinforced our conclusion that Myt1 was a key determinant of AZD1775 sensitivity (Fig. 7k). Notably, HT at least partly decreased Myt1, CDK1-pY15, and CDK1-pT14, augmented HT-induced DNA damage and apoptosis (Fig. 7l), and restored the sensitivity to AZD1775 even in ID8R cells (Fig. 7m), supporting HT has the potential to overcome acquired WEE1i resistance by downregulating Myt1.

### Hyperthermic intraperitoneal perfusion and WEE1i significantly regress ovarian tumors in vivo

Our in vitro experiments provide interesting findings about the synergistic effects of combination therapy of HT and WEE1 inhibition on ovarian cancer cells. Reliable in vivo pre-clinical models will further bridge the gap between in vitro studies exploring the HIPET mechanism and the potential effects in patients in the context of standardized HIPET therapy. Thus, we developed a miniaturized, streamlined, and reproducible device mimicking the clinical treatment protocol to validate the in vivo efficacy of HIPET with WEE1i on mice. This closed-circuit equipment consisted of a temperature-controlled water bath, a perfusate stock bottle, an inflow/outflow catheter, a roller pump, and a temperature-detecting and displaying system (Fig. 8a).

A mouse model of ovarian cancer with miliary peritoneal spread was established by intraperitoneally injecting a luciferase-transfected murine ID8 ovarian cancer cell line (ID8-luc) into C57BL/6 mice. Once tumors were detected with the In Vivo Imaging System (IVIS), 20 mice were randomly divided into four groups (Fig. 8b). Notably, in our preliminary trials evaluating the tolerance of tumor-bearing C57BL/6 mice to HIPET, we found that a duration of 30 min for HIPET was well-tolerated with minimal complications. However, durations exceeding 30 min resulted in significant health issues, including potential mortality. Fortunately, due to the high sensitivity of ID8 to HT (Fig. 3a), we tested the effects of various durations of the HT stimuli (30, 60, and 90 min) on ID8 cells. Intriguingly, 30 min HT stimulus was sufficient to induce CDK1 activation, DNA damage, apoptosis, and reduce cell viability to levels comparable to those seen in other ovarian cancer cells subjected to 90 min of HT (Fig. S4a–c). Moreover, 30 min HT phenocopied the synergistic lethal effects of AZD1775 in ID8 cells (Fig. S4d–f). So, we treated mice with continuous intraperitoneal lavage with normal saline solution as control or with AZD1775 (60 mg/kg) as the treatment group at 37 °C or 42 °C (four groups, each $n = 5$ mice). After a 24 h postoperative recovery period, the two control groups were administered with the vehicle (2% DMSO + 30% PEG300 + 5% Tween 80 + ddH2O, oral gavage). Simultaneously, the two treatment groups received AZD1775 (60 mg/kg/d, oral gavage, 5 days on, 2 days off) for an additional 21 days, or until the need to terminate the mice due to severe ascites in the control groups arose (Fig. 8b). After equipment optimization and procedural adjustments, no mice died during the HIPET procedure or perioperative period, and all survived to the study endpoint; subsequently, IVIS was conducted and tumor tissues were harvested and evaluated regarding

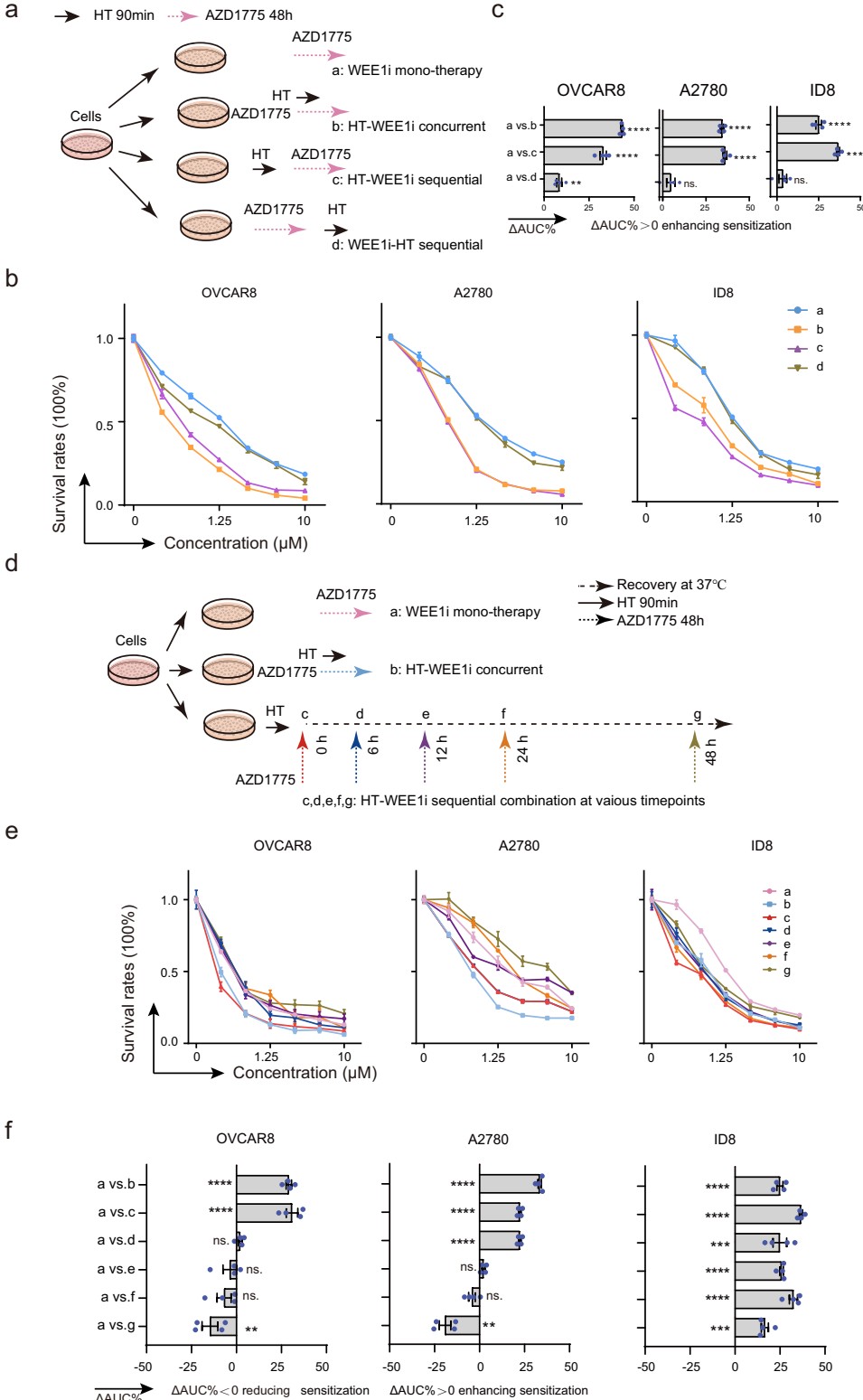

**Fig. 6 | Combined modality strategies and therapeutic windows for the combinational therapy of hyperthermia and WEE1 inhibition. a** Flow diagram of the sequential or concurrent combination modality strategies of hyperthermia (HT, 42 °C for 90 min) and WEE1 inhibitor (WEE1i, AZD1775). **b, c** Dose-response curves of the sequential or concurrent combined strategies of HT and WEE1i in OVCAR8, A2780, and ID8 cell lines. The ΔAUC% was calculated compared to WEE1i mono-therapy (*n* = 4 per group). **d** Flow diagram of the concurrent or various sequential combination strategies with HT and WEE1i. OVCAR8, A2780, and ID8 cells were treated with increasing doses of AZD1775 for 48 h in the presence (concurrent therapy) or absence (WEE1i mono-therapy) of HT incubation or cells were first treated with HT for 90 min, then returned to normothermia (37 °C). AZD1775 was administered at various normothermia recovery time points (0, 6, 12, 24, and 48 h) for 48 h. **e, f** Dose-response curves and ΔAUC% of various combined modality strategies are shown (*n* = 4 per group). Comparisons were performed by unpaired two-tailed Student's *t* test in (**c, f**). Data are presented as mean ± SEM. ****$p$ < 0.0001, ***$p$ < 0.001, **$p$ < 0.01, ns., not significant. All analytical data are derived from a minimum of three biologically independent experiments, with '*n*' indicating the specific number of replicates. Source data are provided as a Source Data file.

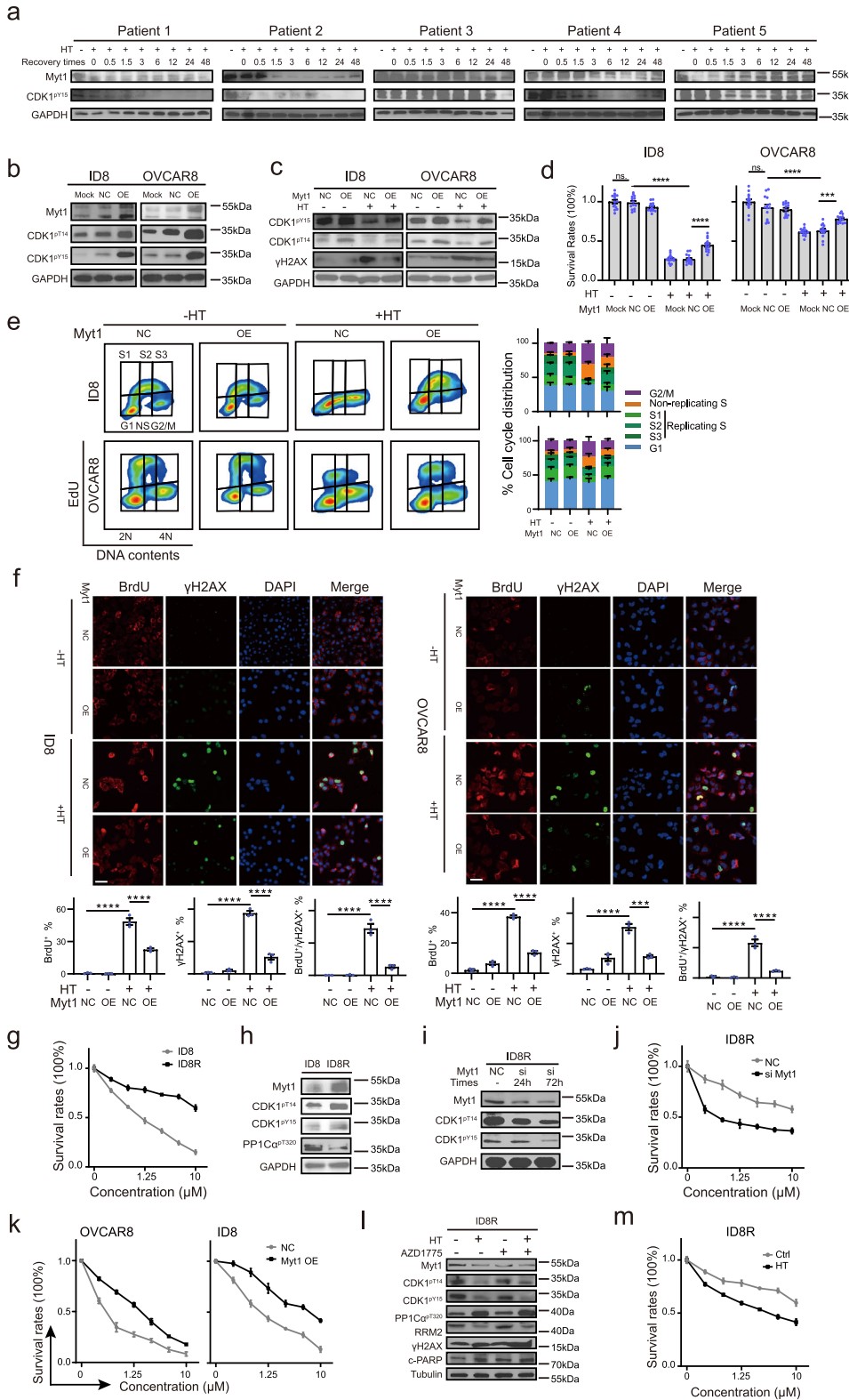

morphology, proliferation, and DNA damage and CDK1 activation by immunohistochemistry (IHC).

Consistent with the high sensitivity of ID8 cells to HT in vitro, a 30 min HT infusion modestly decreased tumor burden as demonstrated by IVIS (Fig. 8c, d), and abdominal circumference indicative of ascites (Fig. 8e), when compared with normothermia-perfused control animals. Meanwhile, AZD1775 mono-therapy achieved a greater tumor burden reduction than HT alone (Fig. 8c–e). Combining WEE1i with HT

via intraperitoneal infusion resulted in significant tumor regression (Fig. 8c–e).

As expected, IHC analysis of tumors at the termination of the study demonstrated that CDK-pY15 was massively decreased in WEE1i-treated tumors, consistent with WEE1i fully inhibiting its target at doses used throughout the duration of the study (Fig. 8f). Notably, the combination of HT and WEE1i further intensified this reduction, suggesting the occurrence of CDK1 hyperactivation in the synergistic

**Fig. 7 | Myt1 downregulation contributed to the hyperthermia-induced CDK1 hyperactivation and the synthetic lethality with WEE1 inhibition.**
**a** Representative western blot results in five primary HGSOC samples across five independent experiments. HGSOC cells were incubated at hyperthermia (HT, 42 °C) for 90 min, then allowed to recover at 37 °C for the indicated times (0, 0.5, 1.5, 3, 6, 12, 24, and 48 h). **b** Representative western blot images of Myt1, CDK1-pY15, and CDK1-pT14 were obtained in ID8 and OVCAR8 cells (Mock) or cells transfected with lentiviral vectors carrying either scramble overexpression (NC) or Myt1 over-expression (OE). **c** Representative western blot images of CDK1-pY15, CDK1-pT14, and γH2AX proteins were obtained in ID8/OVCAR8 cells transfected with NC or Myt1-OE lentiviral vectors after exposure to either 37 °C or 42 °C for 90 min. **d** Cell viability was assessed in ID8/OVCAR8 cells (Mock), NC, and Myt1-OE cells following a 90 min HT exposure ($n = 14$ per group). **e** EdU and PI staining were analyzed by flow cytometry in ID8/OVCAR8 cells transfected with either NC or Myt1-OE and treated as in (**c**) ($n = 3$ per group). **f** Representative images and quantification of BrdU and γH2AX positive staining in ID8/OVCAR8 cells transfected with NC or Myt1-OE and treated as in (**c**) ($n = 3$ per group). Scale bar, 25 μm. **g** Dose-response curves of cell viability in ID8 and ID8R cells treated with increasing doses of AZD1775 for 48 h ($n = 3$ per group). **h** Representative western blot images of the indicated proteins in ID8 and ID8R cells. **i** Representative western blot images of Myt1, CDK1-pY15, and CDK1-pT14 in cells treated as indicated. Cell viability, measured by CCK8, in cells treated as indicated (**j**: $n = 4$ per group; **k**: $n = 3$ per group). **l** Representative western blot images for the indicated proteins were obtained from ID8R cells treated with AZD1775 (800 nM) for 48 h, either alone or in combination with HT. **m** Dose-response curves were plotted for ID8R cells treated as indicated ($n = 3$ per group). Comparisons were performed by one-way ANOVA followed by Tukey's multiple comparisons test in (**d, f**). Data are presented as mean ± SEM. $^{****}p < 0.0001$, $^{***}p < 0.001$, ns., not significant. All analytical data are derived from a minimum of three biologically independent experiments, with "$n$" indicating the specific number of replicates. Source data are provided as a Source Data file.

treatment group (Fig. 8f). Ki67 indicative of proliferation was modestly decreased in tumors with HT and WEE1i mono-therapy, and massively decreased in mice that received HT/WEE1i combinations. This was accompanied by the induction of γH2AX indicative of DNA damage in HT or WEE1i-treated tumors, particularly in mice that received HT/WEE1i combinations (Fig. 8f). Two mice with large amounts of ascitic fluid in the normothermia-treated control group showed with increased numbers of white blood cells (leukocytosis) and low red blood cell counts (anemia) (Fig. S4g), indicating infection and cachexia. The number of white and red blood cell counts in the other groups remained in the normal range (Fig. S4g). Serum chemistry panels did not reveal changes in ALT, AST, BUN, and CREA levels (Fig. S4g). No severe locoregional or systemic toxicity was observed as indicated by tissue pathologic damage, demonstrating tolerability of the combination (Fig. S4h). Notably, consistent results in immunodeficient NOD mice (Fig. S5a–c) indicate that the synergistic anti-tumor effects of HIPET with WEE1 inhibition aren't reliant on an intact immune system.

To further assess the efficacy of HIPET combined with WEE1i, we compared it with traditional HIPEC (HIPET with DDP and TAX[61–64]) in terms of anti-tumor effects and toxicity (Fig. S6a). Our findings revealed that the normothermic perfusion of DDP + TAX markedly attenuates tumor size and ascites, and the addition of HT to DDP + TAX (HIPEC) did not lead to a significant additional decrease in tumor regression (Fig. S6b, c). Furthermore, relative to normothermic DDP + TAX, the HIPEC regimen did not significantly increase DNA damage (γH2AX) or reduce tumor proliferation (Ki67) (Fig. S6d). However, although the anti-tumor efficacy of HIPET combined with WEE1i is comparable to HIPEC, the latter, particularly due to the constituents DDP and TAX, may potentiate hepatotoxicity, especially under hyperthermic conditions (Fig. S6e).

## Discussion

The recently randomized trial performed by van Driel et al. showed a profound improvement in OS with the addition of cisplatin-based HIPEC to cytoreductive surgery (CRS) in EOCs with PM[5]. However, in the KOV-HIPEC-01 trial, the addition of HIPEC to CRS should not be viewed as beneficial for prolonging advanced EOC patients' PFS and OS[65]. Moreover, a controversial result was reported in the PRODIGE-7 trial which showed no survival benefit but increased morbidity in oxaliplatin-based HIPEC-treated colorectal cancer patients[66]. Thus, HIPEC (chemotherapy-associated HIPET) may be active in certain scenarios but not in others, so further studies into the optimal regimen are urgently needed. To date, many HIPEC regimens have been used worldwide but most are platinum-based (oxaliplatin, cisplatin, and carboplatin) chemotherapeutic agents. The application of molecular targeted therapy in the HIPET context is yet to be fully examined. Delving into the cellular and molecular mechanisms governing the response to the HT component of HIPET is essential for devising targeted precise hyperthermic intraperitoneal therapy (P-HIPET) strategies, aiming to enhance the therapeutic benefits of HIPET.

Upon exposure to HT, signaling networks change in cancer cells that might be exploited for their eradication[67]. We applied a comprehensive multi-omics approach to identify and exploit the vulnerabilities engendered by HT, which might provide proof-of-concept therapeutic options for a rational combination of HT with targeted therapy based on these vulnerabilities. Interestingly, this unbiased trans-omics-based approach identified a unique HT-induced molecular process and demonstrated that rapid changes in protein phosphorylation represented the primary cell response upon HT as opposed to changes in protein or RNA abundance. Importantly, kinase activation profiling revealed the CDK1 kinase to be hyperactivated after HT. Specifically, we validated that CDK1 hyperactivation led to DNA repair stress, dNTP pool depletion, and DNA damage accumulation after HT in ovarian cancer cells. A follow-up candidate drug screen identified WEE1, an inhibitory kinase of CDK1, as a molecular target that synergistically kills cancer cells with HT. Using a murine ovarian cancer and PM model and the in-house developed miniaturized, streamlined, and reproducible device mimicking the clinical HIPEC treatment protocol, proof-of-concept was delivered that the combination of HT with WEE1i leads to dramatic anti-tumor responses. These findings provide additional insights into the consequences of HT and its therapeutic exploitation.

CDK1 controls multiple facets of the cell cycle, including spatio-temporal coordination of replication origin activity and entry to mitosis[35,68]. Excessively activated CDK1 can exacerbate replication stress and cause early mitotic entry to intolerable levels, triggering replication, and mitotic catastrophe[36,68]. Here, we propose a model in which the basis for the observed synthetic lethality between HT and WEE1 inhibition is the result of a two-stage CDK1 activation, in which HT-driven Myt1 downregulation and CDK1 hyperactivation render cells vulnerable to the loss of WEE1-driven inhibitory CDK1 Tyr15 phosphorylation. Both WEE1 and Myt1 kinases phosphorylate and inhibit CDK1[59]. Whereas Myt1, localized in the cytoplasm, phosphorylates CDK1 on Thr14 and Tyr15[69], WEE1, which is nuclear-localized, phosphorylates CDK1 at the Tyr15 residue to inhibit these kinases[70]. The concurrent inhibition of Myt1 and WEE1 resulted in the super-activation of CDK1 causing unscheduled replication arrest and mitotic entry that are associated with catastrophic genome instability. Notably, inhibiting other negative regulators of CDK1 such as CHK1i and ATRi during drug screening did not completely replicate the synergism with HT. WEE1 inhibition exacerbates replication stress to evoke massive DNA damage, which may help improve cancer therapy strategies. This difference could be, at least partly, explained by the fact that while WEE1 directly inhibits CDK1 through phosphorylation on CDK1 residues, ATR/CHK1 regulate CDK1 by phosphorylating and inactivating the CDC25 phosphatases which reverse the

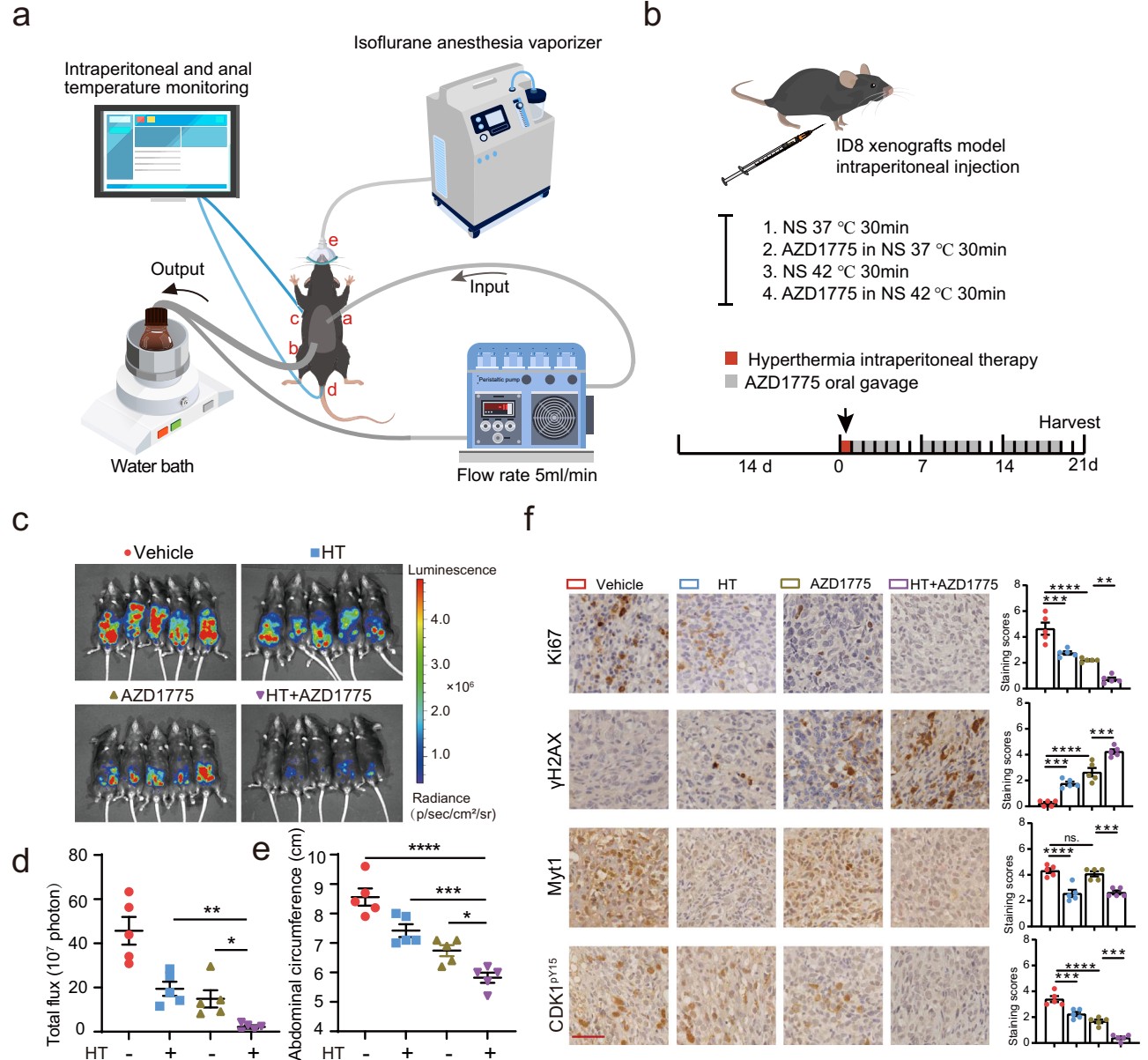

**Fig. 8 | Hyperthermic intraperitoneal perfusion and WEE1i significantly regresses ovarian tumors in vivo. a** A schematic representation of the hyperthermic intraperitoneal therapy (HIPET) setup in mice. The perfusate was heated up and circulated continuously into the abdominal cavity, through an inlet catheter (**a**) into the right and an outlet catheter into the left lower abdomen (**b**). The intraperitoneal (**c**) and anal (**d**) temperatures were constantly monitored. Mice were anesthetized with an isoflurane anesthesia vaporizer (**e**). Third-party elements were adapted from the Freepik material library (https://www.freepik.com/), with hyperlinks included in the Source Data to properly attribute the source. **b** Schema of the mouse experimental protocol. Third-party elements were modified from the Scidraw (https://scidraw.io/), licensed under a Creative Commons Attribution 4.0 Generic License (https://creativecommons.org/licenses/by/4.0/). Representative images (**c**) and quantification (**d**) of bioluminescence in C57BL/6 mice treated as described in (**b**) ($n = 5$ mice per group). **e** Quantification and statistical analysis of the abdominal circumference of mice ($n = 5$ per group). **f** Tissue sections were stained with Ki67, γH2AX, Myt1, and CDK1-pY15. Scale bar, 50 μm. The quantification of IHC staining scores is shown in the right panel ($n = 5$ per group). Comparisons were performed by one-way ANOVA followed by Dunnett's multiple comparisons test in (**d, e**) and one-way ANOVA followed by Tukey's multiple comparisons test in (**f**). Data are presented as mean ± SEM. $****p < 0.0001$, $***p < 0.001$, $**p < 0.01$, $*p < 0.05$, ns., not significant. All analytical data are derived from a minimum of three biologically independent experiments, with "$n$" indicating the specific number of replicates. Source data are provided as a Source Data file.

phosphorylation of T14 and Y15 of CDK1[56,71]. Consistent with this concept, Elbaek et al. reported that CHK1i failed to substantially increase CDK1 activity compared with WEE1i likely because of the more indirect CHK1 mode of achieving CDKs hypo-phosphorylation via phosphatase suppression[72].

Cancer cells exhibit high levels of endogenous replicative stress[73,74], which is a critical factor for the effectiveness of therapeutic interventions targeting DNA damage repair pathways, such as PARP or WEE1 inhibitors[48,75]. In theory, the combinational therapy with HT and WEE1i can further exacerbate this endogenous replicative stress by interfering with exogenous DNA damage responses, such as CDK1-dependent excessive DNA replication stress and DNA damage[76]. So, the pronounced synergistic anti-tumor effect observed in ovarian cancers treated with HT and WEE1 inhibition may arise from the inherent high replicative stress present in these cancer cells. However, it is important to emphasize that the synergistic anti-EOC effects of HT

and WEE1 inhibition are not completely dependent on specific genetic or histological contexts[23,24], although the magnitude of the effect may vary (Fig. S3c). These observations suggest that the synergistic effect observed may be related to the acute and impactful nature of HT as a stimulus, which rapidly and significantly induces high CDK1 activity, subsequently leading to replication catastrophe and mitotic catastrophe.

The major limitations in translating the results of in vitro cell cultures about HIPET to the clinic are the absence of a tumor microenvironment and the lack of systemic phenomena including hydrodynamics, pharmacokinetics, and pharmacodynamics. Thus, clinically relevant in vivo HIPET models are essential for the successful translation of laboratory findings into clinical protocols. While pigs, rats, and rabbits have been widely used as preclinical models to study HIPET procedures[77–79], the lack of tumor and relevant PM models in these animals hamper their applications in evaluating HIPET effects on tumor size or survival. Mice models are more accessible and genetically manipulatable which is suitable for evaluating anticancer activity of HIPEC[80]. Here, we developed a feasible and state-of-the-art HIPECT setup in mice to mimic clinical HIPET conditions, including an anti-visceral tissues adhesion outlet device, a circulation machine with an adjustable flow rate, and a real-time monitoring and regulating system for abdominal, and core temperature. This design provides us an opportunity to investigate the impact of HIPET treatment parameters regarding therapeutic drugs or combinations, drug dosage, carrier solution, perfusate volume, temperature, and duration, on treatment outcomes, which might be able to improve our knowledge of how HIPET can best be utilized in humans. Moreover, the ID8 cell line was utilized to assess the therapeutic efficacy of combining HIPET and WEE1i for treating ovarian cancer peritoneal dissemination in mice. However, it is important to acknowledge that ID8 cells display a heightened sensitivity to HT compared to human-derived ovarian cancer cell lines, presumably due to their species origin[81,82]. Consequently, there is an urgent need to validate these findings using human-derived cell lines or patient-derived xenograft models. This validation process necessitates the development of a safer and more tolerable instrument to accurately reproduce hyperthermic perfusion in humans. Furthermore, to bridge the gap between preclinical insights and their actual implementation in clinical practice, it is warranted to execute further clinical trials incorporating determining optimal dosage and timing, evaluating potential side effects, and precisely identifying appropriate patients.

Together, we present a comprehensive multi-omics (transcriptomel, proteome, and phosphoproteome) study in ovarian cancer cells after HT. Multi-tier integrative analyses led to the identification of a vulnerability to HT inhibition, and data-driven drug screening identified WEE1 as a synthetic lethal target with HT in killing tumor cells. Using an in-house developed HIPET device that mimics clinical treatment protocols, we validated that combining WEE1i could provide an attractive therapeutic strategy for HIPET, aiming to transition from traditional chemotherapy-associated HIPET (HIPEC) to an updated era of P-HIPET. Considering the WEE1i AZD1775 (adavosertib) is currently in multiple phase II clinical trials (http://www.clinicaltrials.gov), assessing the HIPET and WEE1i in the clinical setting should therefore be prioritized to maximize patient benefit.

## Methods
### Study approval
The ovarian tumor samples in this study were collected from patients after surgery at Tongji Hospital, Tongji Medical College, Huazhong University of Science and Technology. Our study adheres to all relevant ethical regulations concerning human participant research. The study was approved by the Institutional Review Board of Tongji Hospital (Permit Number: S080). All participants provided their written informed consent, ensuring a clear understanding and voluntary participation. All related mouse study protocols received approval from the Institutional Animal Care and Use Committee of Tongji Hospital (Permit Number: TJH-202203009) and were conducted following the Chinese Council on Animal Care's guidelines for animal care and use.

### Cell lines
All these cell lines, including HOC7 and OVCAR8 from MDACC's Characterized Cell Line Core, ES2, SKOV3, OVCAR3, Caov3, OV90, TOV-112D, and TOV-21G from the American Type Culture Collection (ATCC), and A2780 from Procell Life Science & Technology Co., Ltd., are human ovarian cancer cell lines. ID8, a mouse ovarian cancer line from C57BL/6, was obtained from Professor K. Roby (Department of Anatomy and Cell Biology, University of Kansas, U.S.A.), who amplified, aliquoted, and cryopreserved the cells at an early passage in liquid nitrogen. The CT26 mouse colon carcinoma line, derived from BALB/c mice, was sourced from ATCC, and the MC38 mouse colon adenocarcinoma line from C57BL/6 was acquired from MDACC's Characterized Cell Line Core. The culture media used included RPMI-1640 media supplemented with 10% fetal bovine serum for HOC7, OVCAR8, A2780, OVCAR3, and Caov3 cells. McCoy 5A medium containing 10% fetal bovine serum was utilized for ES2 and SKOV3 cells. OV90, TOV-112D, and TOV-21G cells were cultivated as monolayers in a medium consisting of MCDB 105 (sodium bicarbonate containing 1.5 g/L) and M199 (sodium bicarbonate containing 2.2 g/L) mixed in a 1:1 ratio, supplemented with 15% fetal bovine serum. ID8, MC38, and CT26 were cultured in DMEM medium with 10% fetal bovine serum. HaCaT cells, a human keratinocyte line, and HUVEC cells, derived from the umbilical cord vein, were both obtained from ATCC and grown in DMEM medium with 10% fetal bovine serum. Additionally, IOSE80 cells, sourced from Cellosaurus, are an ovarian surface epithelium cell line cultivated in the same medium. All cell lines were maintained in a humidified incubator at 37 °C with 5% $CO_2$. Before experimentation, each cell line underwent STR profile analysis authentication and was verified as free of Mycoplasma contamination. See Supplementary Table 2 for comprehensive cell source details.

### Mild hyperthermia treatment in vitro
The cells were permitted to adhere to the bottom of the culture dishes, flasks, or multi-well plates for a minimum of 12 h. Subsequently, they were incubated in a humidified 5% $CO_2$ atmosphere at either 37 °C (Ctrl) or 42 °C (HT) for durations of 30, 60, or 90 min. Cells were harvested or fixed as indicated in specific experiments.

### Sample preparation for the multi-omics profiling
In total, $5 \times 10^6$ OVCAR8 cells were seeded into T175 culture flasks. After 24 h, the medium was replaced with fresh medium pre-heated to either 37 °C or 42 °C, and the cells were then incubated in a humidified 5% $CO_2$ atmosphere at 37 °C (Ctrl) or 42 °C (HT) for 90 min. Post-incubation, cells were washed thrice with phosphate-buffered saline (PBS) and harvested using cell scrapers. Over $2 \times 10^6$ cells were collected and lysed in TRIzol (Takara, 9109) for RNA extraction. For proteomic and phosphoproteomic analysis, more than $2 \times 10^7$ cells were harvested and immediately frozen in liquid nitrogen for subsequent protein extraction. Three independent biological replicates were prepared for each group.

### RNA-seq
RNA was extracted from OVCAR8 cells using the AllPrep DNA/RNA Mini Kit (QIAGEN, 80204). RNA concentration was quantified using the Qubit (Thermo Fisher Scientific), and its quality was assessed using the Agilent 2100 Bioanalyzer (Agilent Technologies). A sample that passed RNA-seq data quality control and had a minimum RNA integrity number score of 7 was sent for sequencing. Library construction was performed using 2 μg RNA per sample. Briefly, mRNA was purified

from total RNA using poly-T oligo-attached magnetic beads. Fragmentation was carried out using the fragmentation buffer. The first-strand cDNA was synthesized, and the second-strand cDNA synthesis was subsequently performed. The remaining overhangs were converted into blunt ends. After adenylation of the 3′ ends of DNA fragments, adaptors with a hairpin loop structure were ligated. Subsequently, PCR was performed. The resulting libraries were quality-controlled and sequenced on the DNBSEQ-G400 platform to generate 150 bp paired-end reads, according to the manufacturer's instructions.

## Sample preparation for proteomics and phosphoproteomics

**Protein extraction.** Samples were sonicated three times on ice using a high-intensity ultrasonic processor (Scientz) in lysis buffer (8M urea, 1% protease inhibitor cocktail, 1% phosphatase inhibitor cocktail). The remaining debris was removed by centrifugation at 12,000 $g$ at 4 °C for 10 min. Finally, the supernatant was collected and the protein concentration was determined with a BCA kit according to the manufacturer's instructions.

**Trypsin digestion.** For subsequent digestion, the protein solution was slowly added until reaching a final concentration of 20% (m/v) trichloroacetic acid (Sigma-Aldrich) to precipitate protein, and then vortexed for mixing and incubated for 2 h at 4 °C. The precipitate was collected by centrifugation at 4500 $g$ for 5 min at 4 °C. The precipitated protein was washed with pre-cooled acetone three times and dried for 1 min. The protein sample was then redissolved in 200 mM TEAB (Sigma-Aldrich) and ultrasonically dispersed. Trypsin was added at a 1:50 trypsin-to-protein mass ratio for the first digestion overnight. The sample was reduced with 5 mM dithiothreitol (DTT, Sigma-Aldrich) for 30 min at 56 °C and alkylated with 11 mM iodoacetamide (Sigma-Aldrich) for 15 min at room temperature (RT) in the darkness. Finally, the peptides were desalted by a Strata X C18 SPE column (Pheonomenex).

**TMT labeling.** After trypsin digestion, peptides were first dissolved in 0.5M TEAB. Each channel of peptide was labeled with their respective tandem mass tags (TMT) reagent (based on the manufacturer's protocol, Thermo Scientific), and incubated for 2 h at RT. Five microliters of each sample were pooled, desalted, and analyzed by mass spectrometry to check labeling efficiency. After the labeling efficiency check, samples were quenched by adding 5% hydroxylamine. The pooled samples were then desalted with a Strata X C18 SPE column (Phenomenex) and dried by vacuum centrifugation.

**HPLC fractionation.** The sample was fractionated into various fractions by high pH reverse-phase high-performance liquid chromatography (HPLC) using an Agilent 300 Extend C18 column (5 μm particles, 4.6 mm ID, 250 mm length). Briefly, peptides were separated with a gradient of 8% to 32% acetonitrile (pH 9.0) over 60 min into 60 fractions. Then, the peptides were combined into 12 fractions for proteomics and 6 fractions for phosphoproteomics. Subsequently, the peptides were dried by vacuum centrifuging.

**Phosphopeptides enrichment.** Peptide mixtures were first incubated with the immobilized metal-ion affinity chromatography (IMAC) microspheres suspension with vibration in a loading buffer (50% acetonitrile/6% trifluoroacetic acid). The IMAC microspheres with enriched phosphopeptides were centrifuged and collected. To remove the non-specifically adsorbed peptides, the IMAC microspheres were washed with 50% acetonitrile/6% trifluoroacetic acid and 30% acetonitrile/0.1% trifluoroacetic acid, sequentially three times. To elute the enriched phosphopeptides, an elution buffer containing 10% $NH_4OH$ was added and the enriched phosphopeptides were eluted with vibration. The supernatant containing phosphopeptides was collected and lyophilized for LC-MS/MS analysis.

## LC-MS/MS analysis

**Proteomics analysis.** The tryptic peptides were dissolved in solvent A (0.1% formic acid, 2% acetonitrile in water) and directly loaded onto a homemade reversed-phase analytical column (25 cm length, 75 μm i.d.). The proteomic gradient comprised an increase of solvent B (0.1% formic acid in 90% acetonitrile) from 7% to 25% in 26 min, then from 25% to 38% in 8 min, climbing to 80% in 3 min, and then holding at 80% for the last 3 min, all at a constant flow rate of 500 nL/min on an EASY-nLC 1200 UPLC system (ThermoFisher Scientific). The separated peptides were analyzed using a Q ExactiveTM HF-X (ThermoFisher Scientific) equipped with a nano-electrospray ion source. The electrospray voltage applied was 2.1 kV. The full MS scan resolution was set to 120,000 for a scan range of 400–1600 m/z. The 25 most abundant precursors were then selected for further MS/MS analyses with 30 s dynamic exclusion. The higher energy collision dissociation (HCD) fragmentation was performed at a normalized collision energy (NCE) of 28%. The fragments were detected in the Orbitrap at a resolution of 15,000. The fixed first mass was set to 100 m/z. The automatic gain control (AGC) target was set to 5E4, with an intensity threshold of 2.5E5 and a maximum injection time of 40 ms.

**Phosphoproteomics analysis.** The tryptic peptides were dissolved in solvent A (0.1% formic acid, 2% acetonitrile in water) and directly loaded onto a homemade reversed-phase analytical column (25 cm length, 75 μm i.d.). The phosphoproteomic gradient of solvent B (0.1% formic acid in 90% acetonitrile) was set as follows: from 4% to 20% in 40 min, from 20% to 32% in 12 min, from 32% to 80% in 4 min, and 80% for the last 4 min, at a constant flow rate of 500 nL/min on an EASY-nLC 1200 UPLC system (ThermoFisher Scientific). The separated peptides were analyzed using Q ExactiveTM HF-X (ThermoFisher Scientific) with a nano-electrospray ion source. The electrospray voltage applied was 2.1 kV. The full MS scan resolution was set to 60,000 for a scan range of 350–1400 m/z. The 20 most abundant precursors were then selected for further MS/MS analyses with 15 s dynamic exclusion. The HCD fragmentation was performed at an NCE of 28%. The fragments were detected in the Orbitrap at a resolution of 30,000. The fixed first mass was set to 100 m/z. The AGC target was set to 5E3, with an intensity threshold of 8.3E4 and a maximum injection time of 60 ms.

## Multi-omics data analysis

**RNA-seq analysis.** Raw data were trimmed by Trim Galore (v.0.6.7) with default settings and then aligned to the human genome (hg38) with STAR[83] (v.2.7.10a). The read count was generated by subread[84] (v.2.0.1) with default parameters and the transcripts per million of genes were quantified using RSEM[85] (v.1.3.1). Differential gene expression analysis was performed in R using DESeq2[86]. Genes with an absolute log2FoldChange of greater than one and a $p$ value < 0.05 were counted as differentially expressed genes. The R package, clusterProfiler[87], was used for the enrichment of DEGs. For Gene Ontology enrichment in terms of biological process, we used the simplify function of clusterProfiler[87] to remove redundant enriched terms. DE analysis of transcriptomic data is elaborated in Supplementary Data 1 for further details.

**MS data analysis.** The resulting MS/MS data were processed using the MaxQuant search engine (v.1.6.15.0). Tandem mass spectra were searched against the database *Homo_sapiens*_9606_SP_20201214.fasta (20395 entries) concatenated with a reverse decoy database. Trypsin/P was specified as the cleavage enzyme allowing up to two missing cleavages. The minimum amino acid length was set to seven. The mass tolerance for precursor ions was set to 20 ppm in the first search and 4.5 ppm in the main search, and the mass tolerance for fragment ions was set to 20 ppm. Carbamidomethyl on Cys was specified as a fixed modification, and acetylation on protein N-terminal and oxidation on Met were specified as variable modifications. The FDR was adjusted to

<1%. The relative quantitative value of the protein was calculated and further normalized using the median-centering method. Genes with a missing value in at least 50% of the samples were filtered, and the reserved missing values were imputed by the K-Nearest Neighbor (KNN) method using the R package impute[88] (v.1.70.0). Differentially expressed proteins were identified using the limma (3.54.2) package[89], using log2FoldChange >0.3 (or <−0.3) and a P value < 0.05 as the cut-off. GSEA analyses were performed with the R package clusterProfiler[87]. Differential analysis of quantitative proteomics data is elaborated in Supplementary Data 2 for further details.

**Phosphoproteomic analysis.** Phosphorylated proteomic analyses were carried out by the post-translational modification (PTM) BioLab. The resulting MS/MS data were processed using the Maxquant search engine (v.1.6.15.0). Tandem mass spectra were searched against the *Homo_sapiens_9606_SP_20201214.fasta* (20395 entries) concatenated with a reverse decoy database. Trypsin/P was specified as the cleavage enzyme, allowing up to two missing cleavages. The mass tolerance for precursor ions was set to 20 ppm in the First search and 4.5 ppm in the Main search, and the mass tolerance for fragment ions was set to 20 ppm. Carbamidomethyl on Cys was specified as a fixed modification, while oxidation of methionine, acetylation of protein N-terminus, deamidation (NQ), and phosphorylation of serine, threonine, and tyrosine were defined as variable modifications. The quantitative method was set to TMT-6plex. The FDR was adjusted to <1%. Genes present in at least 50% of the samples were reserved, and the missing values of preprocessed data were imputed by the KNN method using the R package impute[88]. Differential phosphorylation sites were identified using the limma package[89] with an absolute log2FoldChange >0.5 and a P value < 0.05. Differential analysis of phosphoproteomic data is elaborated in Supplementary Data 3 for further details.

**PTM-SEA analysis.** PTM signatures database (PTMsigDB, v.1.9.0) was first downloaded from http://prot-shiny-vm.broadinstitute.org:3838/ptmsigdb-app/. The PTM-signature enrichment analysis (PTM-SEA) on GitHub (https://github.com/broadinstitute/ssGSEA2.0) was performed using the following parameters. sample.norm.type: rank; weight: 0.75; statistic: area.under.RES; output.score.type: NES; nperm: 1000; min.-overlap: 5; correl.type: z.score.

**The construction of the PPI network.** The protein-protein interaction network of the candidate genes was constructed using the Search Tool for the Retrieval of Interacting Genes database[90] (https://string-db.org/) and visualized using Cytoscape[91] (v.3.1.2).

**Isolation and culture of primary ovarian cancer cells.** Fresh tissue samples were surgically excised and stored in MACS tissue storage solution (Miltenyi Biotec, 130-100-008) until processed. Initially, samples were washed with PBS and finely minced into 1 mm cubic pieces on ice. These pieces were then enzymatically digested using the Tumor Dissociation Kit, human (Miltenyi Biotec 130-095-929), following specifications for the gentleMACS™ Octo Dissociator to obtain single cells. Post-digestion, samples were filtered through a 70 μm cell strainer (Corning, 352350) and centrifuged at 300–500 g for 8 min. The supernatant was discarded, and the cell pellet was resuspended in red blood cell lysis buffer (Servicebio, G2015-500ML) for a 15 min incubation to lyse red blood cells. Cells were then washed with DMEM medium and resuspended in the same medium before being re-filtered through a 40 μm cell strainer (Corning, 352340). The specimen was subsequently collected under sterile conditions in 15 mL tubes. For ascites, 50 mL of fluid was centrifuged, resuspended in 7 mL of red blood cell lysis solution, and incubated for 15 min at 37 °C. This was followed by centrifugation at 300 g for 5 min, discarding the supernatant, and washing the cells thrice to remove most red blood cells. The dissociated cells were then incubated overnight in DMEM + 15% fetal bovine serum in a tissue culture incubator (37 °C, 5% $CO_2$, 95% air). The following day, the medium was removed, and the cell layer was rinsed thrice with 1X PBS to eliminate suspended immune cells. In isolating EOC cells from solid tumors, partial trypsinization (gradient digestions) was employed to selectively detach cells with poorer adhesion, particularly fibroblasts.

**Chemical compounds.** Chemical compounds including AZD1775, THZ1 2HCl, TWS119, PD-0332991, RO-3306, PD0166285, LJI308, ON-01910, LY2606368, MK-2206 2HCl, AMN-107, VE-821, KU55933, KU-57788, NSC 119875, and NSC 125973 were acquired from Selleck. Protein kinase inhibitor 1 hydrochloride was sourced from MCE. Supplementary Table 3 provides IC50 values for 15 inhibitors obtained from cell-free assays, including information on their target specificity, potential secondary targets, as well as their commercial names and catalog numbers. To supply nucleotide precursors, a mix of four dNTPs (50 uM each) was added for 48 h. The dNTP set from Sigma-Aldrich (DNTP100-1KT) consists of four separate 0.25 mL tubes, each containing 100 mM aqueous solutions of deoxyribonucleotides: deoxyribose adenosine triphosphate, deoxyribose cytidine triphosphate, deoxyribose guanosine triphosphate, and deoxyribose thymidine triphosphate.

**CRISPR-Cas9 mediated gene editing in cells.** CRISPR/Cas9 systems were employed for the knockout of human HSF1, TP53, and mouse TP53. SgRNAs, synthesized by TsingKe Co., Ltd., were cloned into the pLentiCRISPR v2 vector using restriction endonucleases BsmBIv2 (NEBiolabs, R0739S) and T4 DNA Ligase (Takara, 2011A). The ligation products were transformed into DH-5α competent bacteria, with all constructs verified by DNA sequencing. For establishing stable cell lines, the pCMV-dR8.91 lentiviral packaging plasmid, pCMV-VSVG envelope plasmid, and sgRNA-pLentiCRISPR v2 constructs were co-transfected into HEK293T cells. These cells, at 80% confluence in 10 cm Petri dishes, were transfected using Lipofectamine 3000 Transfection Reagent (Thermo Fisher Scientific, L3000015). The transfection mixture for each dish included 18.75 μL of Lipofectamine 3000, 25 μL of P3000 Reagent, 4.2 μg each of mutation constructs, pCMV-dR8.91 plasmid, and pCMV-VSVG plasmid. Following a 15 min incubation at RT, the mixture was added to the cells. Virus supernatants were harvested 48 h post-transfection and filtered through a 0.45 μm polyethersulfone filter. For the SKOV3, A2780, and ID8 cell lines, each at 30–40% confluence, cells were treated with 2 mL of virus suspension and 1 mL of complete medium containing 10 μg/mL polybrene (YEA-SEN, 40804ES76). Puromycin screening at 2 μg/mL was initiated 48 h post-infection. After ~4 weeks of selection, stable clones were isolated. Gene knockouts were confirmed using Western blot analyses and DNA sequencing. The sequences of the sgRNAs used are detailed in Supplementary Table 4.

**Stable over-expression cell generation.** The full-length cDNA of the human PKMYT1 gene (NM_004203.5) and the mouse PKMYT1 gene (NM_023058.3) were cloned into the pLVX-C-FLAG-mCMV-ZsGreen-IRES-Puro vector. DesignGene Co., Ltd. carried out the synthesis, and the specific details of plasmid construction can be found in Supplementary Table 5. Stable cell lines were generated via lentivirus transduction and subsequent puromycin selection. Briefly, $6–8 \times 10^6$ HEK293T cells were seeded onto each 10 cm dish. After 24 h, at 80–90% confluency, a mixture of 500 μL solution I (containing 10 μL of 1.0 ug/μL Vector, 15 μL of 1.0 ug/μL Lentiviral packaging mixture, and 50 μL CPT Buffer B dissolved in ddH2O) and 500 μL CPT Buffer A was incubated for 30 min at RT and then added to the HEK293T cells for transfection. Cell media were collected and replaced with fresh media twice, every 24 h, -18 h after transfection. The collected media were pooled, and filtered through a 0.45 μM PVDF filter to remove HEK293T cell debris, and the virus

was harvested. The centrifuged virus was resuspended in PBS buffer and stored at −80 °C.

Post-titration, the lentivirus concentration was established at $1 \times 10^8$ TU/mL. For infection, cancer cells at 30–50% confluency in a six well plate were incubated with a medium mixture of 200 μL lentivirus, 2 μL polybrene (10 mg/mL), and 2.2 mL fresh complete media for 24 h, followed by maintenance in fresh complete media for an additional 24 h. Approximately 48 h post-infection, cells were subjected to media containing an optimal concentration of puromycin for selection (OVCAR8: 2 μg/mL puromycin, ID8: 2.5 μg/mL puromycin). The cells typically attained stability within 48–72 h after selection and were validated by Western blot. The Lentiviral Packaging Mix (NCC200563ZP) and CPT Efficient transfection kit (R001) were sourced from Creative Biolabs and VIRATHERAPY TECHNOLOGIES, respectively.

**RNA interference.** All siRNAs were obtained from Guangzhou RiboBio Co., Ltd. For transfecting cells, all siRNAs were used at a concentration of 100 μM (unless otherwise indicated) with Lipofectamine 3000 Transfection Reagent (Thermo Fisher Scientific, L3000015) for 24 h prior to treatments. Knockdown efficiency was analyzed by Western blot and normalized to GAPDH ~48 h after transfection. The siRNA sequences used are listed in Supplementary Table 6.

**Generation of WEE1 inhibitor-resistant clones.** To develop Wee1 inhibitor-resistant ID8 clones, we subjected the cells to a gradual increase in AZD1775 concentrations over 3 months, reaching a final concentration of 5 μM. Following this exposure, the cells were cultured without AZD1775 for a minimum of 1 month before being utilized in experiments.

**Clonogenic assay.** Five hundred cells per well were seeded in a six-well plate and incubated overnight. The following day, the cells were subjected to incubation at either 37 °C or 42 °C for 90 min, with or without the presence of drugs. After a 2-week incubation period, the remaining cells were fixed with 4% paraformaldehyde, stained with a solution of crystal violet (Sigma-Aldrich, 548-62-9) at RT for 10 min, and then washed with PBS. Images of the colonies were captured using a digital scanner after drying, followed by colony number counting using Image J (version 1.53k).

**Cell proliferation.** Cell proliferation was assessed using the Cell Counting Kit-8 (CCK-8, Domino Laboratories, CK04-11) following the manufacturer's instructions. Optical density values were measured at a wavelength of 450 nm using the SpectraMax ABS Plus Microplate Reader. The area under the fitted dose-response curve (AUC) was calculated to predict drug sensitivity. Sensitization of HT to drugs was quantified as ΔAUC%, calculated as [AUC (Ctrl)−AUC (HT)]/AUC (Ctrl).

**Flow cytometry for apoptosis.** After treatment, cells were harvested and then stained with Annexin V and propidium iodide (PI) using the FITC Annexin V Apoptosis Detection Kit I (BD Biosciences, 556547). The samples were analyzed on a Beckman Coulter CytoFLEX Flow Cytometer, and flow cytometry data were collected using CytExpert (version 2.4). FlowJo V10 software was utilized for the analysis and visualization of the flow cytometry data. The FACS sequential gating strategies were as follows: The starting cell population was selected through FSC/SSC gates, where debris and dead cells with lower forward scatter were excluded. The density plot was segmented into four quadrants to identify cells that were either single-positive for Annexin V-FITC, double-negative, or double-positive for Annexin V-FITC and PI. The relative proportions of early and late apoptosis cells were quantified by placing gates around these distinct populations. Apoptosis levels were quantified by calculating the fractional difference in Annexin V-FITC positive (early) and Annexin V-FITC/PI double-positive

(advanced) populations between treated and untreated samples. At least $1 \times 10^4$ cells were collected to determine the percentage of apoptotic cells.

**Flow cytometry for cell cycle analysis with EdU incorporation and γH2AX detection.** The EdU-647 Cell Proliferation Assay Kit (Beyotime, C0081S) was used according to the manufacturer's instructions. Briefly, cells were incubated in a medium with a final concentration of 10 μM EdU for 2 h, then harvested for fixation in 4% polyformaldehyde for 15 min. After washing with PBS, cells were permeabilized with 0.3% Triton X-100 for 10 min and blocked using 3% bovine serum albumin (BSA) in PBS. The cells were then labeled with Azide 647 click addition solution at RT for 30 min in the dark, and subsequently rinsed with PBS. Cells were then incubated with specific antibodies against γH2AX (CST, 80312, 1:200) for 90 min at RT. Next, cells were washed with PBS and incubated with a secondary antibody conjugated to AlexaFluor 488 (Jackson Immunoresearch, 715-585-151, 1:200) or Pacific Blue (Thermo Fisher, P31582, 1:200) for 1 h. Finally, cells were stained with PI which contained 100 mg/mL RNase A (BD Biosciences, 550825). Flow cytometry data from samples analyzed on a Beckman Coulter Cyto-FLEX Flow Cytometer were collected using CytExpert (version 2.4) and analyzed and visualized with FlowJo V10 software. Information on antibody dilution ratios, manufacturers, and catalog numbers can be found in Supplementary Table 7. A visual representation of the specific FACS sequential gating strategy can be found in Supplementary Fig. 7. The initial cell population was chosen based on FSC/SSC gates, excluding debris and dead cells with lower forward scatter. Gating on single cells and excluding clumps or doublets on a plot of PE-A (forward scatter area) vs PE-H (forward scatter height). In the APC (EdU-Azide 647) versus PE (DNA contents-PI) plot, five distinct groups are displayed, representing cells in S1-phase, S2-phase, S3-phase, G1-phase, Non-replicating S-phase, and G2/M-phase. In the FITC (γH2AX-AlexaFluor 488) or PB450 (γH2AX- Pacific Blue) versus PE plot, γH2AX-positive cells are depicted relative to the negative control. Within the γH2AX-positive cell population, the APC versus PE axis can be used to visualize the distribution of the five cell cycle phases: S1-phase, S2-phase, S3-phase, G1-phase, Non-replicating S-phase, and G2/M-phase.

**DNA fiber assay.** Cells were labeled for 30 min in a medium containing 25 μM CldU (Sigma-Aldrich, CAS:50-90-8), followed by 30 min in a medium containing 250 μM IdU (Sigma-Aldrich, CAS:54-42-2). After treatment, cells were harvested and immediately resuspended in PBS at a concentration of $5 \times 10^5$ cells/mL. 2 μL of the suspension was pipetted onto silane-coated microscope slides, allowed to dry, then 7 μL of the spreading buffer (200 mM *Tris* HCl, pH 7.4, 50 mM EDTA, 0.5% SDS) was added and mixed. To achieve more extended DNA fibers, the DNA was allowed to run down the slide slowly (with slides held at a 70° angle), was air-dried, and then fixed in methanol/acetic acid (3:1) for 10 min. The slides were washed with water, denatured in 2N HCl for 1 h, and then washed with PBS. They were incubated in blocking solution (PBS containing 1% BSA and 0.1% Tween 20) for 1 h, followed by incubation with Rat anti-BrdU/CldU (clone ICR1, Abcam, ab6326, 1:300) antibody and Mouse anti-BrdU/IdU (clone B44, BD Biosciences, 347580, 1:50) antibody for 2 h at RT. After rinsing with PBS, the slides were further incubated with anti-Rat AlexaFluor 488 antibody (Jackson Immunoresearch, 712-545-153, 1:150) and anti-Mouse AlexaFluor 594 antibody (Jackson Immunoresearch, 715-585-151, 1:150) for 1 h at RT. After a final wash, the slides were mounted using the Operetta CLS High-Content Analysis System (PerkinElmer), and image data were collected with Harmony (version 2.9) software. Information on antibody dilution ratios, manufacturers, and catalog numbers can be found in Supplementary Table 7.

**Immunofluorescence staining for γH2AX and pH3.** Cells were seeded onto 96-well cell culture plates and allowed to attach overnight

before exposure to treatments. Following treatment, cells were fixed with 4% paraformaldehyde, permeabilized with 0.5% Triton X-100, blocked with 5% BSA, and subsequently incubated with primary antibodies (anti-phospho-histone H3 Ser10, Abcam, ab5176, 1:200; anti-γH2AX, CST, 80312, 1:200) overnight at 4 °C. This was followed by incubation with secondary antibodies (anti-Rabbit AlexaFluor 488 antibody, Jackson Immunoresearch, 711-545-152, 1:200; anti-Mouse AlexaFluor 594 antibody, Jackson Immunoresearch, 715-585-151, 1:150, 1:200) for 1 h at RT. Nuclei were counterstained with DAPI (1 mg/mL, Servicebio, G1012). Cells were observed and imaged under the ImageXpress Micro Confocal High-Content Analysis System (Molecular Devices). Cell nuclei and positive cells were quantified using ImageJ software (version 1.53k). Information on antibody dilution ratios, manufacturers, and catalog numbers can be found in Supplementary Table 7.

**Detection of nuclear-derived ssDNA and γH2AX detection.** Briefly, cells were labeled with 10 μM BrdU (Sigma-Aldrich, CAS:59-14-3) for 48 h before being subjected to the indicated treatments (in the absence of BrdU). Cells were then fixed with 4% polyformaldehyde for 15 min. BrdU detection under non-denaturing conditions was performed using a specific rat monoclonal antibody (anti-BrdU [clone BU1/75 (ICR1)], Abcam, ab6326, 1:200), while γH2AX was detected using rabbit anti-γH2AX antibodies (ABclonal, AP0687, 1:200) overnight at 4 °C. After rinsing, cells were incubated with anti-Rat Alexa-Fluor 594 antibody (Jackson Immunoresearch, 712-585-153, 1:200) and anti-Rabbit AlexaFluor 488 antibody (Jackson Immunoresearch, 711-545-152, 1:200) for 1 h at RT. Finally, cells were stained with DAPI and imaged using an ImageXpress Micro Confocal High-Content Analysis System (Molecular Devices). Cell nuclei and positive cells were quantified using ImageJ software (version 1.53k). Information on antibody dilution ratios, manufacturers, and catalog numbers can be found in Supplementary Table 7.

**Western blot.** Cells were washed once with PBS and lysed in RIPA buffer (Servicebio, G2002) containing EDTA-free Protease Inhibitor Cocktail (Servicebio, G2006-250UL) and phosphatase inhibitor (Servicebio, G2007). Protein concentration was measured using Coomassie dye (Beyotime, ST1119,). Ten percent SDS-polyacrylamide gel electrophoresis was performed to separate 20 μg of total protein, which was subsequently transferred onto a nitrocellulose membrane (Bio-Rad). Membranes were blocked with 5% BSA and then incubated with the specified primary antibodies at 4 °C overnight, followed by incubation with HRP-conjugated secondary antibodies (ABclonal, AS014, 1:5000) for 1 h at RT. Enhanced chemiluminescence (Advansta, K-12045-D50) was used to detect the specific blot bands using the ChemiDoc Imaging System (Bio-Rad). Blot bands were processed using Image Lab software (version 6.0.1). Information on antibody dilution ratios, manufacturers, and catalog numbers can be found in Supplementary Table 7.

**IHC staining.** Tissues were fixed overnight in 4% polyformaldehyde and subsequently embedded in paraffin. Following deparaffinization, rehydration, antigen unmasking, and endogenous peroxidase blocking, sections were blocked for 60 min in 5% BSA with 0.1% Triton X-100 in PBS. Subsequently, tissue sections were stained using a Mouse/Rabbit Enhanced Polymer Method Detection System (ZSGB-BIO, PV-9000). Slides were recognized and images were captured using the Slide Scan System SQS-600P 40X/80X (Teksqray). Tumor cell staining was scored using a semi-quantitative five-category grading system while staining intensity was assessed with a semi-quantitative four-category grading system. Staining scores were calculated as the average of the tumor cell staining score multiplied by the staining intensity score. Information on antibody dilution ratios, manufacturers, and catalog numbers can be found in Supplementary Table 7.

**Establishment of intraperitoneal metastatic ovarian tumors in mice.** Female C57BL/6 and NOD mice (6–7 weeks old) were procured from Gempharmatech Co., Ltd. The animals were maintained in sterile conditions at a temperature of 20–25 °C with 50% humidity and a 12 h light-dark cycle. Cages, covers, bedding, food, and water were changed and sterilized every week. A period of 2 weeks was allotted for the acclimatization of all animals before commencing the experiment. The mice received an intraperitoneal injection of an ID8 cell suspension ($1 \times 10^7$ cells in 100 μL of PBS media for C57BL/6 and $5 \times 10^6$ cells in 100 μL of PBS media for NOD mice). To ensure even cell distribution, the abdomen of the injected animals was gently massaged. Two weeks after injection, the mice had developed multiple small disseminated intraperitoneal tumors measuring 1–2 mm in size. All procedures were performed under sterile conditions.

**HIPEC procedure.** A mouse model of EOC with miliary spreads was established by intraperitoneally injecting luciferase-transfected murine ID8 ovarian cancer cell lines (ID8-luc) into C57BL/6 mice. Tumor detection was performed using the IVIS (PerkinElmer IVIS Lumina III System). Tumor-bearing mice were anesthetized with 2.5% isoflurane in 100% oxygen using an isoflurane anesthesia vaporizer, and anesthesia was maintained with 1% isoflurane throughout the entire procedure. To prevent heat loss, mice were placed in a supine position on a heating pad set to 37 °C (extremity fixation was not required). The peritoneal perfusate, consisting of 100 mL of normal saline solution with or without drugs, was warmed to 53 °C by immersing the perfusate stock bottles into a thermostatically-controlled water bath, taking ~15 min. The infusion was driven by a roller pump at a rate of 5 mL/min. Two catheters, the inflow (a) and outflow (b) catheters, were introduced into the left hypochondrium and right iliac fossa of the abdominal wall, respectively (as shown in Fig. 8a). The inflow catheter was adapted from an indwelling needle, while the outflow apparatus was designed in-house. In small animal models, ensuring unobstructed flow throughout the HIPEC procedure is a challenge, as visceral and fat tissues can easily obstruct the outflow catheter. To address this, we designed a reticular construction that effectively separates the outflow region from the peritoneal cavity, ensuring unobstructed drainage without negative pressure.

After a few seconds of infusion, the abdomen bulged sufficiently to facilitate perfusate drainage and gentle abdominal massage was applied to achieve uniform perfusate distribution. The intraperitoneal temperature was measured in real-time using nickel-chrome-nickel thermocouples placed in the middle of the abdominal cavity and displayed with a multi-channel data-logging digital thermometer during the procedure. The infusion flow rate was intelligently adjusted by controlling the roller pump to optimize thermal homogeneity around 42 °C (±0.5 °C) during HIPEC treatments. Additionally, core temperature was monitored using a rectal thermocouple. If the core temperature exceeded 38.5 °C, the heating mat was removed, and the body temperature was cooled by placing the tail in 70% ethanol.

Perfusion was conducted for 30 min after reaching the required temperature. After perfusion, the inflow catheter was removed, and light pressure on the abdomen was applied to extract surplus solution before removing the perfusate outflow catheter. Finally, the abdominal wall was closed with single sutures and disinfected with betadine. Mice were placed on warm mats and monitored for recovery and discomfort.

**Experimental design in vivo.** Two weeks post-tumor cell injection, C57BL/6, and NOD mice were randomly grouped (C57BL/6: $n = 5$, NOD: $n = 3$ per group) and underwent HIPET (with or without AZD1775). In the control groups, intraperitoneal infusion of normal saline solution (37 °C or 42 °C) was carried out for 30 min. In the drug treatment groups, animals were treated with intraperitoneal AZD1775 for 30 min at 37 °C or 42 °C. The infusion tube was connected to a brown bottle

containing 100 mL normal saline solution with AZD1775 at a dose of 60 mg/kg body weight. The dose of AZD1775 used for infusion was determined based on oral dosing in mice[48,49]. After a 24 h post-operative recovery period, the two control groups received a vehicle (2% DMSO + 30% PEG300 + 5% Tween 80 + ddH2O, 100 µL oral gavage), while the two drug treatment groups received AZD1775 (dissolved in 2% DMSO + 30% PEG300 + 5% Tween 80 + ddH2O, 60 mg/kg/day, oral gavage, 5 days on, 2 days off). Fluorescence images were taken 21 days after drug administration to monitor residual tumors (Fig. 8b). Tumor tissue and blood samples were obtained after sacrificing the mice for further analysis.

For the HIPEC procedure, C57BL/6 mice were also randomly divided into four groups ($n = 5$ per group). In the control groups, intraperitoneal infusion of normal saline solution (37 °C or 42 °C) was conducted for 30 min. In the drug treatment groups, animals were treated with intraperitoneal DDP and TAX for 30 min at 37 °C or 42 °C. The infusion tube was connected to a brown bottle containing 100 mL of normal saline solution with both DDP (5 mg/kg) and TAX (10 mg/kg). After 1 week, the two control groups received a vehicle (100 µL PBS, intraperitoneal). In contrast, the two treatment groups were treated with DDP (5 mg/kg) and TAX (10 mg/kg) (intraperitoneal, per week) for another 2 weeks. Fluorescence images were taken 21 days after HIPEC to monitor residual tumors (Fig. S6a), and tumor tissues were harvested and evaluated by IHC.

Following institutional ethics committee guidelines, cases involving ascites formation due to peritoneal tumor dissemination are managed according to the prescribed protocol. Rodents are euthanized when their body weight exceeds 120% of the initial weight. Additionally, in the context of the multiple tumor models for ovarian cancer, it is stipulated that multiple tumors, each smaller than the single tumor limit, may not have the same negative impact as a single large tumor. Therefore, multiple tumors are permitted to grow up to a total diameter of 3.0 cm in mice, in accordance with the specified guidelines. The study ensures adherence to ethical guidelines regarding maximal tumor size/burden.

### Statistics and reproducibility

The legends of the figures define the number of biological replicates. The statistical data presented are derived from at least three biologically independent experimental replicates, and the results are expressed as mean ± SEM. Statistical analysis and data plotting were conducted using GraphPad Prism (version 9.0.0) software. To account for the limited sample size, normality was assessed by examining histograms. Comparisons were performed using unpaired Student's two-tailed $t$ test or one-way analysis of variance followed by Dunnett's or Tukey's multiple comparisons test. The figure legends offer detailed descriptions of the specific statistical methods used. Statistically significant differences were considered for $p$ values less than 0.05. In the figures, asterisks denote significance levels as $*p < 0.05$, $**p < 0.01$, $***p < 0.001$, $****p < 0.0001$, and ns. indicate non-significant results. The Source Data file contains more detailed and comprehensive statistical results.

### Reporting summary

Further information on research design is available in the Nature Portfolio Reporting Summary linked to this article.

## Data availability

All mass spectrometry proteomics and phosphoproteomics data have been deposited to the ProteomeXchange Consortium via the PRIDE partner repository with the dataset identifier PXD040849. RNA-seq data are available in the Gene Expression Omnibus (GEO) database under accession code GSE227393. PTM signatures database (PTMsigDB, v1.9.0) was downloaded from https://proteomics.broadapps.org/ptmsigdb/. Any additional information required to reanalyze the data presented in this paper is available from the lead contact upon request. Source data are provided in this paper.

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

## Acknowledgements
This work was supported by the National Natural Science Foundation of China (82373218 to C.S., 82073259 to G.C., 82303671 to R.X., 82060472 to F.Li), the National Key R&D Program of China (2022YFC2704200, 2022YFC2704202 to C.S.), the Key R&D Program of Hubei Province (2023BCB023 to C.S.), and the Fundamental Research Funds for the Central Universities (2042023kf0084 to R.X.).

## Author contributions
The design, execution, validation of the experiments, and data analysis were carried out by X.Y., X.H., J.Y., and W.L., with Y.F., B.Y., and J.F. contributing additional support and assistance. F.Lu, T.Q., X.K., X.Z., R.X., and J.Y. collected and verified the data. T.S., J.L., F.Li, and K.S. helped perform the analysis with constructive discussions. X.Y., X.H., and C.S. wrote the manuscript. C.S. and G.C. revised the manuscript. C.S., G.C., J.L., and X.Y. conceived the project. C.S., G.C., and J.L. performed the supervision and project administration.

## Competing interests
The authors declare no competing interests.

## Additional information

[1]Cancer Biology Research Center (Key Laboratory of the Ministry of Education), Tongji Hospital, Tongji Medical College, Huazhong University of Science and Technology, Wuhan 430000 Hubei, PR China. [2]Department of Gynecology and Obstetrics, Tongji Hospital, Tongji Medical College, Huazhong University of Science and Technology, Wuhan 430000 Hubei, PR China. [3]Department of Obstetrics and Gynecology, Qilu Hospital of Shandong University, Jinan 250012, PR China. [4]Department of Obstetrics and Gynecology, The First Affiliated Hospital of Shihezi University Shihezi, Xinjiang 832000, PR China. [5]Department of Gynecology and Obstetrics, Zhongnan Hospital of Wuhan University, Wuhan 430071 Hubei, PR China. [6]Ovarian Cancer Program, Department of Gynecologic Oncology, Zhongshan Hospital, Fudan University, Shanghai 200032, PR China. [7]Department of Gynecologic Oncology, Sun Yat-sen Memorial Hospital, 33 Yingfeng Road, Guangzhou 510000, PR China. [8]These authors contributed equally: Xiaohang Yang, Xingyuan Hu, Jingjing Yin, Wenting Li.
✉e-mail: lijing228@mail.sysu.edu.cn; tjchengang@hust.edu.cn; suncydoctor@gmail.com

