## [Peer Review File · Nature Communications]

REVIEWER COMMENTS

Reviewer #1 (Remarks to the Author): Expert in MS-based proteomics and phosphoproteomics, and ovarian cancer

The manuscript “Comprehensive multi-omics analysis revealed WEE1 as a synergistic lethal target with hyperthermia through CDK1 super-activation” uses a multi-omic approach, including transcriptomics, proteomics, and phosphoproteomics on human OVCAR8 ovarian cancer cells exposed to 42°C for 90 minutes to determine the molecular effects of hyperthermia as a first step to understanding the variable results obtained in clinical trials combining inter-peritoneal hyperthermia with platinum chemotherapy. The multi-omic analyses are very well done, with biological triplicates and appropriate controls. Unsurprisingly, the authors demonstrate that classical RNA signatures of heat exposure are not manifested at the protein level over a 90 minute time span, and that the most dynamic response to heat treatment within 90 minutes is in protein phosphorylation. The authors then use the proteomic and phosphoproteomic results to identify CDK1 signaling as uniquely sensitive to hyperthermia (HT), demonstrate the downstream effects of CDK1 activation on replication stress, cell cycle transit, DNA damage, and cell death. Using a combination of proteomic data and curated literature guided drug screens, the authors identify WEE1 and Myt1 as potent modifiers of the CDK1 response which might be leveraged to increase the effectiveness of HT as an adjuvant therapy. Data are provided demonstrating synthetic lethality between HT-induced CDK1 activation and WEE1 inhibition with AZD1775, a small molecule inhibitor approved for use in clinical trials, and intriguing preliminary results of efficacy are presented using a syngeneic mouse model of ID8 cells in C57/BL6 mice and an in-house developed device that delivers clinical style hyperthermia to mice.

This manuscript has many significant strengths but a few areas that do need to be addressed. The manuscript is clearly written and generally well organized, with just a few typos that affect meaning [such as ‘includes for induces(?) on line 368 and HGSCO for HGSOC on line 401]. A major strength of the manuscript is the use of multiple cell lines, with biological triplicates, to avoid the common pitfall of basing biological conclusions on a single cell line. Experiments in long-established cell lines are further validated in primary cultures from patient tumors, reducing the possible artifacts due to adaptation to 2D culture. The Methods section is adequate for replication of the experiments and has adequate assurances of quality control. Figure legends are generally appropriate for understanding the figure without constant reference to the main text. Most of the significant scientific conclusions have been supported by orthogonal measurements, such as Western blots to support the proteomic measurements, and multiple distinct assays for DNA damage, apoptosis, and cell cycle transit. The manuscript tells a highly relevant biological story that has the potential to inform clinical practice, specifically the transient, reversible nature of HT effects on specific targets, the role of protein phosphorylation in identifying potential therapeutic targets, and the importance of timing combination therapies to exploit the biology of the underlying system.

That said, the manuscript has one consistent concerning aspect that needs to be addressed in the writing. The ID8 cell line is substantially different from the OVCAR8 and A2780 cell lines that form much of the basis for the molecular pathways depicted in Figures 1 through 6. As acknowledged by the

authors, the ID8 line is uniquely sensitive to the effects of HT, and the effects of HT are more persistent in the ID8 cells than in OVCAR8 or A2780 cells (Figures 1E, 2A, 3A, and most significantly, 6B and 6E), yet nowhere do the authors discuss the possible reasons for these differences. OVCAR8 and A2780 cells are derived from patient HGSOc tumors and have been cultured for decades. Whatever the cell type of origin of HGSOc is (i.e., the controversy over fallopian tube epithelium (FTE) versus ovarian surface epithelium (OSE)), it is represented by these patient derived cell lines. The ID8 cells were derived from spontaneous malignant transformation of mouse ovarian surface epithelial cells, and they are highly valued for their utility in syngeneic mouse models. However, even syngeneic mouse models are an imperfect model for the human patient, with substantial differences in the immune system, plus we do not know if there are inherent differences in OSE versus FTE that might affect molecular interactions such as synthetic lethality. I am not saying that the ID8 mouse model is irrelevant; I am saying that these issues need to be openly discussed in the Discussion section as potential limitations of the study. Furthermore, in this particular study, the most convincing pre-clinical evidence for a possible clinical trial comes from the one cell line that was most sensitive to the effects of HT.

The manuscript narrative, particularly the Discussion, is also disappointing in the lack of any discussion of the interaction between DNA damage repair systems and the effects of replicative stress, particularly given the known importance of the DNA damage repair system in human cancers of the breast and ovary. Moreover, there is significant evidence of increased replicative stress in naïve HGSOc, compared to normal FTE (such as PMIDs 32029455 and 32529193). The thesis of the authors would actually be strengthened by referencing more of the prior literature on replicative stress in HGSOc.

Reviewer #2 (Remarks to the Author): Expert in ovarian cancer clinical research, hyperthermia therapy, and mouse models

Yang et al., manuscript titled “Comprehensive multi-omics analysis revealed WEE1 as a synergistic lethal target with hyperthermia through CDK1 super-activation” focused on identifying the mechanisms of action of hyperthermic intraperitoneal chemotherapy (HIPEC) in epithelial ovarian cancer. Using a comprehensive set of analytical approaches including transcriptomic, proteomic, and phosphoproteomic analyses in several ovarian cancer lines exposed to normothermia or hyperthermia at 42C for 90 minutes. The investigators identified hyperthermia induced gene signature by transcriptomic analysis and more interestingly revealed CDK1 dependent cell cycle checkpoint was the most significantly activated. The findings indicate that CDK1 is hyperactivated by hyperthermia. CDK1 activity was found to be reversible with subsequent replication arrest and early mitotic entry in ovarian cancer cells after hyperthermia. Moreover, WEE1 inhibits CDK1 activity and WEE1 inhibitors induce stress and cell death via CDK1 dependent pathways. Thus a combination of hyperthermia plus WEE1 inhibition leads to synthetic lethality. This is further supported by a preclinical mouse study combining hyperthermia and WEE1 inhibitor.

The studies are comprehensive and offer a new mechanism being activated by hyperthermia. The identification of CDK1 as a signaling node in response to hyperthermia is well supported including the mechanisms of stalled fork DNA replication. Inhibition by CDK1 further supports the finding of this signaling node. Of note, the studies focus on 3 cell lines, OVCAR8, A2780, and mouse ID8. While the cells provide validation in cell culture there is no evidence presented that CDK1 can be activated in patients by HIPEC. This is a major limitation as the investigators propose their studies provide insights on HIPEC.

The findings are narrow providing and identify new insights on hyperthermia activation of cancer cells that is not activated in non-tumor cells. The treatment is strictly hyperthermia and this is far different from HIPEC. The abstract, introduction, and discussion focus on these studies having impact on HIPEC and none of the results support this conclusion. As such this manuscript is limited in scope and is most informative for hyperthermia activation of cancer cells so the application to the clinic is not apparent.

The choice of cancer cells is likewise unclear. Most of the studies use OVCAR8 and A2780, lines that are not representative high grade serous ovarian cancer, the predominant ovarian cancer subtype. The studies do provide some limited replication in patient derived ovarian cancer cells. This data is quite limited with no information on the tumor grade, response to chemotherapy, and outcomes (Fig. 4J and K). Patients 4 and 5 appear to show very different response to hyperthermia and AZD compared to patients 1,2, and 3 in Fig. 4K.

The availability of the ID8 mouse line and ability to study the impact of hyperthermia is quite valuable. Is the efficacy dependent on an intact immune system?

The findings do not consider the impact of the tumor microenvironment including the immune infiltrating cells. The problem is that the studies focus only on the cancer cells which may not be the target of HIPEC. A thorough analysis of the tumors could yield insights on the response of the tumors to hyperthermia and combination with WEE1i.

Additional considerations.

The manuscript aims to provide insights into how to improve HIPEC in EOC, however the mechanisms of WEE1 inhibition are in combination with hyperthermia alone, in the absence of chemotherapeutics. The investigators cannot claim a mechanism into HIPEC improvement without the addition of chemotherapy, as it does not model the mechanisms of HIPEC.

How does CDK1 get activated by hyperthermia? Is activation heat shock dependent or independent?The HIPEC procedure outlined in the methods (line 931) describes a heated peritoneal perfusate containing

normal saline with or without the WEE1 inhibitor. The elegant model needs to be compared to the efficacy of chemotherapy used for HIPEC. How long does the hyperthermia + WEE1i last? There is no mention of HIPEC procedure success, did all mice survive to study endpoint?

HIPEC patients receive 90 minutes of heated chemotherapy – why was 30 minutes used in this study? need to clarify.

Upon harvesting tumors, no analysis of tumor tissue was conducted. Why? The IHC analysis is insufficient.

There are grammar errors throughout the manuscript, including the abstract. Please get an editor to review.

Reviewer #3 (Remarks to the Author): Expert in ovarian cancer genomics and therapy

The study focuses on the effects of combined hyperthermia and WEE1 inhibition in ovarian cancer treatment. The authors present a well-designed study with multiple experiments to assess the efficacy of this combination strategy in ovarian cancer cell lines and mouse models.

Overall, the results of this study are noteworthy and may have a significant impact on the field. The authors aim to demonstrate that the combination of hyperthermia and WEE1 inhibition results in increased DNA damage and cell death compared to either treatment alone in several ovarian cancer cell lines. The methodology is comprehensive, covering multiple cell lines and in vivo models, as well as diverse assays to evaluate the impact of the combination therapy on cell viability, cell cycle, and DNA damage.

However, the writing quality throughout the manuscript is suboptimal, with several instances of unclear or confusing language. The manuscript would benefit from thorough proofreading and editing for clarity, grammar, and punctuation.

1. Make sure to clearly define all abbreviations when they are first introduced, for example, EdU, γ H2AX, and CCK8.

2. Consider reorganizing the paragraph structure to better highlight the key findings and their implications.
3. When introducing the concept of the 'one-two punch' approach, provide a brief explanation of the strategy and its relevance to the study. This will help readers who may not be familiar with the term to better understand the rationale for the drug screening and combination therapy (line 289).
4. The statement "sequential combination of HT and WEE1 inhibitor only occurred before 3, 6, and more than 24 hours in OVCAR8, A2780, and ID8 cells, respectively" could be clearer. It is suggested to rephrase it for better clarity, e.g., "the synergistic lethal effects of sequential combination of HT and WEE1 inhibitor were observed when HT was applied before WEE1 inhibition, with optimal therapeutic windows of 3 hours for OVCAR8 cells, 6 hours for A2780 cells, and more than 24 hours for ID8 cells" (line 380-394).
5. There are some instances where the language could be improved for better readability. For example, the sentence "Meanwhile, Myt1 overexpression blunted HT-induced DNA damage (reflected by γ H2AX accumulation) and cell death in ID8 and OVCAR8 cells" could be rephrased as "Furthermore, overexpression of Myt1 diminished hyperthermia-induced DNA damage (as indicated by reduced γ H2AX accumulation) and cell death in ID8 and OVCAR8 cells" (line 418-420).
6. When presenting the results, consider using a more concise approach. For example, instead of stating "Strikingly, the addition of WEE1i to hyperthermia delivered via intraperitoneal infusion led to profound tumor regression (Fig. 8C-E)," you could say "Remarkably, combining WEE1i with hyperthermia via intraperitoneal infusion resulted in significant tumor regression (Fig. 8C-E)."
7. Some sentences appear to have typos or errors, such as "...and incubated with and incubated with primary antibody..." and "Cells were subsequently contained with DAPI...". These errors should be corrected to maintain the professional tone of the paper.
8. In the HIPEC procedure subsection, the term "emerging" should be replaced with "immersing" when describing warming the perfusate stock bottles.

Methodology:

1. Provide more context for the choice of cell lines and justify the selection of the specific hyperthermia conditions (e.g., 42°C for 90 minutes) and WEE1 inhibitor concentrations.
2. Describe in more detail the experimental procedures, including the number of replicates, control conditions, and any potential confounding factors considered.
3. Include details about the antibodies used for immunoblotting and immunohistochemistry, such as dilution ratios, manufacturers, and catalog numbers.
4. Justify the use of the specific mouse model and discuss any potential limitations or caveats associated with this model.

5. In the IHC staining subsection, the authors should specify the duration of blocking in 5% BSA with 0.1% Triton X-100 in PBS.
6. In the establishment of intraperitoneal metastatic ovarian tumors in mice subsection, the authors should provide the number of mice used for this part of the study.
7. Provide clearer explanations for some parts of the methods, such as the rationale behind the choice of AZD1775 dosages and administration schedules for the in-vivo experiments.
8. In the experimental design in vivo subsection, it is unclear how many days after drug administration the fluorescence images were conducted. The authors should specify this information.
9. In the statistical analysis subsection, the authors should provide information on the normality of the data distribution and the assumptions met for the chosen statistical tests.

Results and data presentation

1. When presenting the results for Figure 1, clarify the mechanism by which hyperthermia leads to increased DNA damage and how this is related to WEE1 inhibition. Please also clarify the relationship between the endogenous DNA damage response, cell cycle phase and the response to exogenous DNA damage.
2. When presenting the results of the drug screening, consider providing more information on the specific inhibitors tested and their targets. This will give readers a better understanding of the scope of the screening and the potential mechanisms of action of the inhibitors.
3. It would be helpful to specify the cell lines used in the experiments for this section, as this information is not provided (line 349 -368).
4. The authors should provide a brief rationale for exploring the different combination strategies (sequential or concurrent). This would help readers understand the significance of these experiments.

Discussion and conclusions:

1. Explore potential off-target effects of the WEE1 inhibitor or other factors that could influence the response to hyperthermia and discuss how these may have impacted the study outcomes.
2. Elaborate on potential limitations of the study, such as the use of cell lines and mouse models, which may not fully represent the complexity of human ovarian cancer.
3. Discuss the clinical relevance of the findings and any potential challenges or considerations for translating the results into clinical practice.

In summary, the manuscript presents a valuable contribution to the field of ovarian cancer research, with compelling evidence supporting the potential of combining hyperthermia and WEE1 inhibition as a therapeutic strategy. However, the manuscript requires significant revisions to address the concerns

mentioned above, particularly in terms of writing quality and clarity. I recommend that the authors revise and resubmit their manuscript to address these concerns before it can be considered for publication.

Point-to-Point Response

We have carefully studied each comment and suggestion and have taken them into account while revising the manuscript. With great respect for your expertise, we present the revised version of our paper, which reflects the efforts we have made to address the issues raised during the review process.

Response to Reviewer #1 (Remarks to the Author)

The manuscript “Comprehensive multi-omics analysis revealed WEE1 as a synergistic lethal target with hyperthermia through CDK1 super-activation” uses a multi-omic approach, including transcriptomics, proteomics, and phosphoproteomics on human OVCAR8 ovarian cancer cells exposed to 42°C for 90 minutes to determine the molecular effects of hyperthermia as a first step to understanding the variable results obtained in clinical trials combining inter-peritoneal hyperthermia with platinum chemotherapy. The multi-omic analyses are very well done, with biological triplicates and appropriate controls. Unsurprisingly, the authors demonstrate that classical RNA signatures of hyperthermia exposure are not manifested at the protein level over a 90 minute time span, and that the most dynamic response to hyperthermia within 90 minutes is in protein phosphorylation. The authors then use the proteomic and phosphoproteomic results to identify CDK1 signaling as uniquely sensitive to hyperthermia (HT), demonstrate the downstream effects of CDK1 activation on replication stress, cell cycle transit, DNA damage, and cell death. Using a combination of proteomic data and curated literature guided drug screens, the authors identify WEE1 and Myt1 as potent modifiers of the CDK1 response which might be leveraged to increase the effectiveness of HT as an adjuvant therapy. Data are provided demonstrating synthetic lethality between HT-induced CDK1 activation and WEE1 inhibition with AZD1775, a small molecule inhibitor approved for use in clinical trials,

and intriguing preliminary results of efficacy are presented using a syngeneic mouse model of ID8 cells in C57/BL6 mice and an in-house developed device that delivers clinical style hyperthermia to mice.

This manuscript has many significant strengths but a few areas that do need to be addressed. The manuscript is clearly written and generally well organized, with just a few typos that affect meaning [such as 'incudes for induces(?) on line 368 and HGSCO for HGSOC on line 401]. A major strength of the manuscript is the use of multiple cell lines, with biological triplicates, to avoid the common pitfall of basing biological conclusions on a single cell line. Experiments in long-established cell lines are further validated in primary cultures from patient tumors, reducing the possible artifacts due to adaptation to 2D culture. The Methods section is adequate for replication of the experiments and has adequate assurances of quality control. Figure legends are generally appropriate for understanding the figure without constant reference to the main text. Most of the significant scientific conclusions have been supported by orthogonal measurements, such as Western blots to support the proteomic measurements, and multiple distinct assays for DNA damage, apoptosis, and cell cycle transit. The manuscript tells a highly relevant biological story that has the potential to inform clinical practice, specifically the transient, reversible nature of HT effects on specific targets, the role of protein phosphorylation in identifying potential therapeutic targets, and the importance of timing combination therapies to exploit the biology of the underlying system.

Response: Thank you for your positive evaluation of our manuscript. We appreciate your attention to detail and identification of a few typos. We have made the necessary corrections based on your suggestions. Specifically, we have changed "incudes" to "induces" in line 368 and "HGSCO" to "HGSOC" in line 401. Additionally, we have thoroughly reviewed the entire manuscript to ensure that similar mistakes have been eliminated.

That said, the manuscript has one consistent concerning aspect that needs to be addressed in the writing. The ID8 cell line is substantially different from the OVCAR8

and A2780 cell lines that form much of the basis for the molecular pathways depicted in Figures 1 through 6. As acknowledged by the authors, the ID8 line is uniquely sensitive to the effects of HT, and the effects of HT are more persistent in the ID8 cells than in OVCAR8 or A2780 cells (Figures 1E, 2A, 3A, and most significantly, 6B and 6E), yet nowhere do the authors discuss the possible reasons for these differences. OVCAR8 and A2780 cells are derived from patient HGSOC tumors and have been cultured for decades. Whatever the cell type of origin of HGSOC is (i.e., the controversy over fallopian tube epithelium (FTE) versus ovarian surface epithelium (OSE)), it is represented by these patient derived cell lines. The ID8 cells were derived from spontaneous malignant transformation of mouse ovarian surface epithelial cells, and they are highly valued for their utility in syngeneic mouse models. However, even syngeneic mouse models are an imperfect model for the human patient, with substantial differences in the immune system, plus we do not know if there are inherent differences in OSE versus FTE that might affect molecular interactions such as synthetic lethality. I am not saying that the ID8 mouse model is irrelevant; I am saying that these issues need to be openly discussed in the Discussion section as potential limitations of the study. Furthermore, in this study, the most convincing pre-clinical evidence for a possible clinical trial comes from the one cell line that was most sensitive to the effects of HT.

Response: Thank you for raising the issue of the significant differences in the thermo-sensitivity between murine-derived ID8 cells and human-derived cell lines (such as OVCAR8 and A2780). Several factors may contribute to this disparity:

- 1) **Species origin:** In line with our findings, numerous studies on hyperthermia have shown that murine cell lines exhibit higher sensitivity to hyperthermia compared to human immortalized cell lines (PMID:28638734, PMID: 1737369, PMID: 561655, PMID:19845488). Adkins et al. demonstrated that the murine colorectal cancer cell line CT26 appeared more sensitive to hyperthermia (42°C, 1h) compared to human tumor cells (OV90 and A549) (PMID: 28638734). Moreover, Morrissey et al. demonstrated that murine cell line C3H 10T (1/2) showed a more rapid and dramatic response to heat stress compared to human glioma cells

(PMID:19845488).

- 2) Tissue origin: As you mentioned, OVCAR8 and A2780 are derived from patient HGSOE tumors, which might originate from fallopian tube epithelium (FTE) or ovarian surface epithelium (OSE). On the other hand, ID8 cells were derived from spontaneous malignant transformation of mouse ovarian surface epithelial cells. However, based on our *in vitro* experiments involving 11 different ovarian cancer cell lines, including various pathological subtypes, we observed that while there were variations in the degree of sensitivity, they all fell within a similar range (figure S3C). Additionally, Barnes et al. (PMID: 34470661) utilized distinct transcriptional programs to categorize immortalized ovarian cancer cell lines into five primary histological subtypes. Specifically, A2780 and TOV112D fall under the endometrioid subtype, OVCAR8, CAOV3 and OVCAR3 are classified as serous, SKOV3 and TOV21G as clear cell, and OV90 as mucinous. Evidently, regardless of the pathological subtype, all human ovarian cancer cells exhibited synthetic lethality between hyperthermia and WEE1 inhibition (figure S3C). Therefore, although we cannot completely rule out the influence of cell origin, we have reason to believe that the origin of tumor cells is not the sole determinant of thermo-sensitivity. Additionally, different pathological subtypes tend to benefit to some extent from combination therapies.
- 3) Genomic or molecular features of tumor cells: For instance, Hatakeyama et al. demonstrated varied hyperthermia sensitivities in ovarian and uterine tumors and identified CTGF as a central regulator of this sensitivity (PMID: 27806300). Moreover, Kanamori et al. suggested that the ubiquitin pathway was more active in heat-resistant SKOV3 cells compared to heat-sensitive A2780 cells (PMID: 34282188). This differential sensitivity likely arises from disparities in cell cycle regulation, DNA damage responses, DNA replication stress, and metabolic status in distinct cells.

We have added these information in discussion of our revised manuscript. (Lines 625-628)

Moreover, we acknowledge that relying solely on findings from murine models may have inherent limitations when extrapolating to human contexts. Actually, we have made several attempts to establish an ovarian cancer peritoneal dissemination model using human ovarian cancer cells, specifically OVCAR8, in Nude mice. However, we encountered significant challenges due to the lower heat tolerance observed in Nude mice compared to C57 mice. These mice were unable to tolerate the 90-minute hyperthermic perfusion treatment, leading to complications such as pulmonary edema and subsequent mortality. This limitation is indeed a current constraint of our study. In light of these limitations, we are actively exploring alternative approaches to validate our findings using safer hyperthermal therapy which could be conducted in Nude mice with human-derived ovarian cancer cells, and even patient-derived xenograft (PDX) models. These efforts aim to provide a more accurate and clinically relevant understanding of hyperthermic therapy in the context of human ovarian cancer. According to your kind suggestion, these limitations have been articulated in the discussion section of our manuscript. (Lines 628-631)

The manuscript narrative, particularly the Discussion, is also disappointing in the lack of any discussion of the interaction between DNA damage repair systems and the effects of replicative stress, particularly given the known importance of the DNA damage repair system in human cancers of the breast and ovary. Moreover, there is significant evidence of increased replicative stress in naïve HGSOC, compared to normal FTE (such as PMIDs 32029455 and 32529193). The thesis of the authors would actually be strengthened by referencing more of the prior literature on replicative stress in HGSOC.

Response: Thank you for your valuable suggestion. As you mentioned, existing literature supports the notion of increased replicative stress in naïve HGSOC compared to normal fallopian tube epithelium (FTE) (PMIDs 32029455 and 32529193). Moreover, recent studies have highlighted the importance of endogenous replicative stress in cancer cells which is critical factor for the effectiveness of therapeutic interventions targeting DNA damage repair pathways, such as PARP or WEE1

inhibitors (PMID: 31185210, PMID: 28008184). Considering these findings, it is plausible that our results demonstrating a remarkable synergistic anti-tumor effect of hyperthermia and WEE1 inhibition in ovarian cancers may be associated with the inherent replicative stress observed in these cells. The combination of hyperthermia and WEE1 inhibition may further exacerbate the existing replicative stress, leading to enhanced cytotoxicity and tumor growth inhibition. (Lines 593-606)

However, it is important to emphasize that the synergistic anti-EOC effects of hyperthermia and WEE1 inhibition are not completely dependent on specific genetic or histological contexts, although the magnitude of the effect may vary (Fig. S3c). For instance, the A2780 ovarian cancer cell line, characterized by TP53 wild-type and Brca2 mutant-type status, exhibits a similar synergistic effect compared to the TP53 mutant and Brca2 wild-type OVCAR8 cell line. Furthermore, while disruption of the G1 checkpoint through TP53 mutation enhances sensitivity to WEE1 inhibitors, the combined effect does not depend on the TP53 mutation status (new fig. S3f, g). These observations suggest that the synergistic effect observed may be related to the acute and impactful nature of hyperthermia as a stimulus, which rapidly and significantly induces high cyclin-dependent kinase (CDK) activity, subsequently leading to replication catastrophe and mitotic catastrophe. In response to your comments, we have integrated these discussions into the main text to offer a comprehensive perspective on the topic. (Lines 321-330, 601-606)

Fig S3, related to Fig 4

f Western blot analysis of P53, CDK1-pY15, and γH2AX proteins in Scramble-ko (NC) or CRISPR-Cas9 mediated TP53-ko clones of A2780 and ID8 cells after being exposed to either 37 °C or 42 °C for 90 min. *g* Dose-response curves for mono-AZD1775 or concurrent HT- AZD1775 therapy in Scramble-ko (NC) or TP53-ko A2780/ID8 cells treated with graded concentrations for 48 h.

Response to Reviewer #2 (Remarks to the Author)

Yang et al., manuscript titled “Comprehensive multi-omics analysis revealed WEE1 as a synergistic lethal target with hyperthermia through CDK1 super-activation” focused on identifying the mechanisms of action of hyperthermic intraperitoneal chemotherapy (HIPEC) in epithelial ovarian cancer. Using a comprehensive set of analytical approaches including transcriptomic, proteomic, and phosphoproteomic analyses in several ovarian cancer lines exposed to normothermia or hyperthermia at 42C for 90 minutes. The investigators identified hyperthermia induced gene signature by transcriptomic analysis and more interestingly revealed CDK1 dependent cell cycle checkpoint was the most significantly activated. The findings indicate that CDK1 is hyperactivated by hyperthermia. CDK1 activity was found to be reversible with subsequent replication arrest and early mitotic entry in ovarian cancer cells after hyperthermia. Moreover, WEE1 inhibits CDK1 activity and WEE1 inhibitors induce stress and cell death via CDK1 dependent pathways. Thus, a combination of hyperthermia plus WEE1 inhibition leads to synthetic lethality. This is further supported by a preclinical mouse study combining hyperthermia and WEE1 inhibitor.

The studies are comprehensive and offer a new mechanism being activated by hyperthermia. The identification of CDK1 as a signaling node in response to hyperthermia is well supported including the mechanisms of stalled fork DNA replication. Inhibition by CDK1 further supports the finding of this signaling node. Of note, the studies focus on 3 cell lines, OVCAR8, A2780, and mouse ID8. While the cells provide validation in cell culture there is no evidence presented that CDK1 can be activated in patients by HIPEC. This is a major limitation as the investigators propose their studies provide insights on HIPEC.

Response: We greatly appreciate the reviewer’s recognition that our study offers a comprehensive exploration of the mechanisms activated by hyperthermia in the context of epithelial ovarian cancer. We deeply apologize for the inadvertent confusion caused by conflating the concepts of hyperthermic intraperitoneal chemotherapy (HIPEC) and hyperthermic intraperitoneal therapy (HIPET) in our manuscript. Previous clinical trials HIPEC have yielded contradictory or suboptimal benefits,

potentially due to an insufficient understanding of the tumor cell changes induced by hyperthermia, leading to a lack of diverse combined therapeutic strategies that can be effectively combined with hyperthermia. Here, we aim to provide new insights for HIPET through comprehensively characterizing the molecular mechanism of hyperthermia, intending to identify potential combination of HIPET with precision-targeted therapeutic drugs, aspiring to transition from the traditional chemotherapy-associated HIPET (HIPEC) towards an era of targeted precision treatment. We term this strategy as "Precise Hyperthermic Intraperitoneal Treatment (P-HIPET)." We apologize for any confusion caused by this error and will ensure that the appropriate terminology is used throughout the revised version of the manuscript.

Moreover, we totally agree that validation of CDK1 activation in patients undergoing hyperthermia is important. As our primary objective is to investigate the molecular changes and therapeutic vulnerability induced by hyperthermia, we have made efforts to collect patient samples, specifically the paired ovarian cancer samples before and after isolated hyperthermic intraperitoneal therapy (HIPET) without the use of chemotherapeutic agents, to verify and validate our findings in a clinical setting. However, we regret to find that currently there are no ongoing clinical trials in China specifically evaluating isolated hyperthermic intraperitoneal therapy treatment. Unfortunately, this means that we are unable to validate the CDK1 activation in patients undergoing hyperthermia currently. We understand the significance of such validation and will continue to explore opportunities to conduct a HIPET "windows" clinical trial and then get relevant patient samples for further investigate the molecular changes associated with hyperthermia.

So, according to your suggestion, we've amended our manuscript to clearly emphasize our study's focus on rational design the "Precise Hyperthermic Intraperitoneal Treatment (P-HIPET)". And we also added a discussion emphasizing further investigations that warranted to translate our findings effectively into clinical practice. (Lines 548-551, 640-643)

Notably, following your comments, we tried to acquire some ovarian cancer samples before and after HIPEC with collaboration of Prof. Li in Sun Yat-sen Memorial

Hospital. These pre- and post-HIPEC samples were sourced from a specific single-center clinical study, "Feasibility and safety of neoadjuvant laparoscopic hyperthermic intraperitoneal chemotherapy in patients with advanced stage ovarian cancer: a single-center experience" (PMID: 31440085). In this study, HIPEC was given to patients within 24 hours post-initial laparoscopic assessment. The NACT regimen included intravenous paclitaxel (175 mg/m² over 3 hours) followed by cisplatin (70 mg/m²) via HIPEC. Using Borey Medical's precision device with temperature accuracy of $\pm 0.10^{\circ}\text{C}$ and flow precision of $\pm 5\%$, cisplatin was mixed with 3000 ml saline and circulated at 300-500 ml/min, maintaining 43°C. The 90-minute HIPEC process comprised 30 minutes of preheating and 60 minutes of perfusion. After neoadjuvant laparoscopic hyperthermic intraperitoneal chemotherapy (NLHIPEC), two more NACT cycles were administered every 21 days, followed by interval debulking surgery (IDS) within four weeks post-final NACT cycle. Our samples were derived from both the initial laparoscopic biopsy and final IDS. To delve deeper, we used immunohistochemical staining on these samples, focusing on the CDK1pY15 as a marker. Notably, post one NLHIPEC and two NACTs, the HGSOc samples displayed increased CDK1pY15 when compared to the initial laparoscopic biopsy (see appended figure). This could be due to the DNA damage caused by the platinum drugs in NLHIPEC and NACT, triggering the DDR pathway and activating cell-cycle checkpoints. Regrettably, we couldn't discern the isolated effect of hyperthermia during HIPEC on CDK1 activity from these samples.

IHC staining of CDK1-pY15 in tissue sections from PMID: 31440085. Scale bar is 250 μ m. NLHIPEC: neoadjuvant laparoscopic hyperthermic intraperitoneal chemotherapy; IDS: interval debulking surgery. Only presented in the response letter.

Meanwhile, we believe that the insights gained from our experimental findings and analysis of the primary culture ovarian cancer samples can still provide valuable information regarding the effects of hyperthermia and WEE1 inhibition in ovarian cancer. 1) our study employs multi-omics sequencing techniques, integrating proteomics and phosphoproteomics data. Through kinase-substrate integration analysis, we elucidate the "brittle" change of CDK1 activation. 2) Rigorous validation is conducted across 11 ovarian cell lines, involving extensive molecular functional assays such as Western blot, CDK1-specific substrate detection, and phenotype assays specific to CDK1 activity. This comprehensive approach confirms the activation of CDK1 post hyperthermia, establishing its consistency beyond sporadic findings from individual experiments. Moreover, further validation is carried out using multiple clinical primary culture samples. 3) the developed intraperitoneal hyperthermic perfusion apparatus in mice and the constructed ID8 mouse peritoneal ovarian cancer metastasis model also confirmed increased CDK1 activity post-HIPET. So, the results obtained so far provide preliminary evidence supporting the notion that hyperthermia can induce CDK1 activation, and support the rational of combination of HIPET and WEE1 inhibition.

The findings are narrow providing and identify new insights on hyperthermia activation of cancer cells that is not activated in non-tumor cells. The treatment is strictly hyperthermia and this is far different from HIPEC. The abstract, introduction, and discussion focus on these studies having impact on HIPEC and none of the results support this conclusion. As such this manuscript is limited in scope and is most informative for hyperthermia activation of cancer cells so the application to the clinic is not apparent.

Response: As mentioned in our previous response, we acknowledge the confusion might arise from the conflation of HIPET and HIPEC terminologies in our manuscript. We deeply regret this error and apologize for any misinterpretation it may have caused.

To clarify, in light of the suboptimal benefits of traditional HIPEC, we try to develop a novel strategy, the "Precise Hyperthermic Intraperitoneal Treatment (P-HIPET), rather than simply adding new drug components to improve responsiveness of HIPEC. By focusing on the vulnerabilities generated by hyperthermia, our study presents a pioneering effort to explore new avenues within the field of hyperthermia-based treatments without the inclusion of chemotherapy agents.

In this study, we validated the strategy through extensive in vitro experiments involving ovarian cancer cells and primary tumor tissue samples. Our in vivo experiments, simulating clinical hyperthermic intraperitoneal perfusion equipment and treatment procedures, further support the feasibility of combining HIPET with molecular targeted therapy, suggesting the potential for clinical translation of this "new strategy". Admittedly, we recognize the need for more sophisticated preclinical models, such as the peritoneal metastatic mouse model based on PDX, humanized mice that fully simulate the human immune system, and a more comprehensive surgical protocol complemented by a post-operative care regimen for mice. Furthermore, by collaborating with Prof Li, our teams are also looking forward to launch a "windows" clinical study to discuss the safety and efficacy of HIPET combined with targeted drugs. So, according to your valuable suggestion, we have made the necessary revisions to ensure that our manuscript accurately portrays the limited scope of our findings and the need for future research to bridge the gap between preclinical insights and its translation into clinical practice. (Lines 625-635, 640-643)

The choice of cancer cells is likewise unclear. Most of the studies use OVCAR8 and A2780, lines that are not representative high grade serous ovarian cancer, the predominant ovarian cancer subtype. The studies do provide some limited replication in patient derived ovarian cancer cells. This data is quite limited with no information on the tumor grade, response to chemotherapy, and outcomes (Fig. 4J and K). Patients 4 and 5 appear to show very different response to hyperthermia and AZD compared to patients 1,2, and 3 in Fig. 4K.

Response: As you mentioned, Domcke et al. (PMID: 23839242) delineated ovarian cancer cell line models into classifications of 'likely', 'possibly', and 'unlikely' HGSOC cells. Indeed, A2780 is characterized as an 'unlikely' model for this subtype, given its flat copy-number landscape, intact TP53 status, and non-canonical mutations. OVCAR8 is categorized 'possibly' in the classification by Domcke et al. as OVCAR8 cell line exhibits a prominent molecular characteristic of high-grade serous cancer, namely TP53 gene mutation. Additionally, while our molecular functional experiments primarily utilized the OVCAR8 and A2780 cell lines for mechanistic exploration and phenotypic validation, we verified the isolated hyperthermia-induced CDK1 activation and the synergistic anti-tumor effect of hyperthermia and WEE1 inhibitor across 11 immortalized ovarian cancer cell lines available in our lab, including ID8-murine cell, HOC7, OVCAR8, A2780, OVCAR3, TOV112D, TOV21G, ES2, CAO3, SKOV3, and OV90-human cells. These 11 immortalized cell lines, with diverse mutational backgrounds, potentially represent different ovarian cancer subtypes. Notably, CAO3 and OVCAR3 are categorized as "likely" HGSOC according to Domcke's evaluation. Interestingly, we observed that hyperthermia-induced CDK1 activation in ovarian cancer cells and the synergistic activity of the combination of hyperthermia and WEE1 inhibitor were independent of the mutation status of ARID1A, ATM, ATR, BRCA1/2, PIK3CA, PTEN, and TP53 (Revised Fig. S3c, adding the genetic background profile of various tumor cells). This broader validation across different pathological subtypes and genetic background is a significant aspect of our study and constitutes a major part of our findings. Especially, a study (PMID: 30940866) suggests that ovarian cancer cell lines derived from non-serous (NS) carcinomas exhibit more aggressive migration and invasion than those derived from high-grade serous carcinomas. This finding emphasizes the significance of NS tumors and their increased migration and invasion capabilities. Consequently, despite constituting a relatively small proportion of ovarian cancers, NS tumors must not be disregarded in research due to their elevated invasiveness.

Fig S3, related to Fig 4

c The $\Delta AUC\%$ of AZD1775 alone or concurrent HT-AZD1775 treatment in cancer cells with the indicated genetic background profile. The p values were determined by Student's t -test, and the results are presented as the mean \pm SEM. *** $p < 0.001$, ** $p < 0.01$, * $p < 0.05$.

Furthermore, the most prominent molecular characteristics observed in HGSOC is a remarkably high TP53 mutation rate, reaching approximately 97%. Cells with TP53 mutations are believed to have a defect in the G1/S cell cycle checkpoint. These cells strongly rely on the G2/M checkpoint to halt the cell cycle and repair damaged DNA. Therefore, the use of WEE1 inhibitors to release the G2/M cell checkpoint, aiming for synthetic lethality, is considered to have broad application prospects in the field of ovarian cancer treatment. Our findings demonstrate that hyperthermia alone can induce CDK1 activation in both the TP53 wild-type cell line A2780 and TP53 mutant cell lines such as OVCAR8, CAOV3, and OVCAR3. Furthermore, there's a marked synergistic effect when combining hyperthermia with a WEE1 inhibitor in these cells. Additionally, with CRISPR-Cas9-mediated TP53 knockout (TP53ko) (newly added Fig. S3f), hyperthermia was found to induce CDK1 activation in both TP53-WT parental and

TP53-KO cell lines. We also conducted CCK assays combining hyperthermia with a WEE1i, and the results revealed a synergistic effect in both TP53 -WT and -KO cells (newly added Fig. S3g). This indicates that the functional status of TP53 does not impact the synergistic effect between hyperthermia and WEE1i. These findings also suggest that hyperthermia may aid in broadening the clinical application scope of WEE1i. (Lines 321-330)

Fig S3, related to Fig 4

f Western blot analysis of P53, CDK1-pY15, and γH2AX proteins in Scramble-ko (NC) or CRISPR-Cas9 mediated TP53-ko clones of A2780 and ID8 cells after being exposed to either 37 °C or 42 °C for 90 min. *g* Dose-response curves for mono-AZD1775 or concurrent HT- AZD1775 therapy in Scramble-ko (NC) or TP53-ko A2780/ID8 cells treated with graded concentrations for 48 h.

As pointed out by the reviewer, patient 4 and 5 exhibited a suboptimal response to hyperthermia and the WEE1i, AZD1775 (Fig 4J and K). This observed variance in response underscores the potential heterogeneity in ovarian cancer and further emphasizes the necessity of comprehensive studies involving a larger patient cohort. We have supplemented the information with the tumor grade, chemotherapy response, and post-treatment data for these 5 patients (newly added Supplementary Table 1). Unfortunately, patient #3 didn't receive postoperative chemotherapy after tumor cytoreduction in our hospital, and patient #4, after a laparoscopic biopsy, didn't undergo further treatment. Thus, we couldn't assess their chemotherapy responsiveness and prognosis. The differential response observed in patients 4 and 5 compared to patients 1, 2, and 3 highlights the heterogeneity that can exist within ovarian cancer and further emphasizes the need for comprehensive studies involving a larger patient population. (Line 365-368)

The availability of the ID8 mouse line and ability to study the impact of hyperthermia is quite valuable. Is the efficacy dependent on an intact immune system?

Response: Thanks for your positive recognition of the ID8 mouse model in this study. To address the reviewer's concerns regarding the potential dependency of the synergistic effect between HIPET and WEE1i on an intact immune system, we established an ID8 ovarian cancer peritoneal dissemination model in immunodeficient NOD mice. We applied the same treatment regimen as in the C57 mice. Results indicate that, even in the immunodeficient mice, the combination of HIPET and WEE1i still exerted a synergistic anti-tumor effect by inducing CDK1 hyperactivation (newly added Fig S5c) and effectively suppressing tumor growth (newly added Fig. S5a, b). We have included the new results and revised the manuscript accordingly. (Lines 514-516)

Fig S5, related to Fig 8

a Representative images (left) and quantification of bioluminescence (right) in NOD mice after treatment as in Fig 8b. *b* Quantification and statistical analysis of the abdominal circumference of NOD mice, recorded every 7 days. *c* Tissue sections were stained with Ki67, γH2AX, and CDK1-pY15. Scale bar, 50 μm. The quantification of

IHC staining scores is shown in the right panel. The *p* values were determined by a one-way ANOVA (a, b, c), and the results are presented as the mean \pm SD. *****p* < 0.0001, ****p* < 0.001, ***p* < 0.01, **p* < 0.05.

The findings do not consider the impact of the tumor microenvironment including the immune infiltrating cells. The problem is that the studies focus only on the cancer cells which may not be the target of HIPEC. A thorough analysis of the tumors could yield insights on the response of the tumors to hyperthermia and combination with WEE1i.

Response: We value the reviewer's insight on the tumor microenvironment and immune response. To address this concern, we conducted additional analyses on tumor samples from C57 immunocompetent mice treated with HIPET combined with AZD1775. IHC staining for CD4, CD8, and α -SMA on paraffin-embedded sections revealed no heightened immune infiltrating cells in the combination treatment group compared to the AZD1775 monotherapy. Combined with the results obtained from the aforementioned animal experiments, we preliminarily speculate that the synergistic anti-tumor effects of HIPET and WEE1 inhibition are primarily attributed to CDK1 hyperactivation within tumor cells.

C57 mice treated as in Figure 8b. Tumor tissue sections were stained with CD4, CD8 and α -SMA. Scale bar is 250 μ m. Quantification of IHC staining scores were shown in right panel. The *p* values were determined by a one-way ANOVA, and the results are presented as the mean \pm SD. *****p* < 0.0001, ***p* < 0.01, **p* < 0.05, ns., not significant. Only presented in the response letter.

Additional considerations.

The manuscript aims to provide insights into how to improve HIPEC in EOC, however the mechanisms of WEE1 inhibition are in combination with hyperthermia alone, in the absence of chemotherapeutics. The investigators cannot claim a mechanism into HIPEC improvement without the addition of chemotherapy, as it does not model the mechanisms of HIPEC.

Response: As elucidated above, our manuscript inadvertently conflated the concepts of hyperthermic intraperitoneal chemotherapy (HIPEC) and hyperthermic intraperitoneal therapy (HIPET). We emphasize that our research aim to develop a novel strategy, the "Precise Hyperthermic Intraperitoneal Treatment (P-HIPET), rather than simply adding new drug components to improve responsiveness of HIPEC. Once again, we sincerely thank the reviewer for bringing this issue to our attention. We have taken it seriously and made the necessary modifications to ensure the accurate and clear presentation of our research objectives and scope. (Lines 59-63, 548-551, 640-643)

How does CDK1 get activated by hyperthermia? Is activation heat shock dependent or independent?

Response: In alignment with the reviewer's concerns, several studies (PMID: 34686682, 19158073, and 28360195) have indeed indicated an association between CDK1 and the heat shock response (HSR). HSF1 activates the transcription of heat shock proteins, crucial for cellular protection during stress conditions (PMID: 32297210). To investigate whether the activation of CDK1 after hyperthermia depends on HSR mediated by HSF1, we generated stable HSF1 knockout (HSF1-KO) SKOV3 cell lines using CRISPR-Cas9 technology (newly added Fig. S1n). In our study, we observed that hyperthermia-induced Myt1 downregulation and its associated CDK1 activation in SKOV3 cells are independent of HSR (newly added Fig. S1o). We've integrated this data and made corresponding updates to the manuscript. (Lines 222-226, 449-450)

Fig S1, related to Fig 1

n Western blot analysis of the HSF1 expression in CRISPR-Cas9 mediated HSF1-ko SKOV3 cells. **o** Western blot analysis of HSF1, Myt1, CDK1-pY15, and CDK1-pT14 proteins in Scramble-ko (NC) or HSF1-ko SKOV3 cells after being exposed to either 37 °C or 42 °C for 90 min.

The HIPEC procedure outlined in the methods (line 931) describes a heated peritoneal perfusate containing normal saline with or without the WEE1 inhibitor. The elegant model needs to be compared to the efficacy of chemotherapy used for HIPEC.

Response: As you suggested, to assess the potential of combining HIPET with WEE1 inhibition for ovarian cancer treatment, it would be prudent to conduct comprehensive comparisons with traditional HIPEC in terms of synergistic anti-tumor effects, systemic toxicity, and organ-specific toxicities. So, we further evaluate the synergistic effect of hyperthermia and the combination of cisplatin and paclitaxel in the context of HIPEC with ID8 intraperitoneal metastasis model in C57/BL6 mice. First, we determined the dosing and treatment regimen for mouse HIPEC by referencing previous literature (PMID: 34192990, 29108718, 21696631, and DOI: 10.1101/2021.05.16.444343). To elaborate, we treated mice with continuous intraperitoneal lavage with isotonic saline solution as control or with cisplatin (5mg/kg) and taxol (10mg/kg) as treatment group at 37 °C or 42 °C (4 groups, each n = 5). After 1 week, the two control groups were treated with vehicle (100ul PBS, ip), and the two treatment groups were treated with cisplatin and taxol (5mg/kg/ 10mg/kg, ip, per w) for another 14 days until need to terminate mice due to severe ascites in control groups

(newly added Fig. S6a), then IVIS was conducted and tumor tissues were harvested and evaluated regarding morphology, proliferation, and DNA damage by IHC. Furthermore, we evaluated the therapeutic toxicity of HIPEC strategies (newly added Fig. S6a-e).

The additional experimental results indicate that the normothermic perfusion of DDP+TAX can significantly reduce the tumor size and ascites. However, when comparing the combination of hyperthermia with DDP+TAX (HIPEC) to the single drug group, there is a relative tumor regression, but it is not statistically significant (newly added Fig. S6b. c). Moreover, this combination results in severe liver function abnormalities (newly added Fig. S6e). Furthermore, compared to the normothermic perfusion group with TAX and DDP, the HIPEC group did not result in a more pronounced reduction in DNA damage (γ -H2AX) or tumor invasiveness (Ki67), as seen in newly added Fig S6d. While the combined HIPET and WEE1i regimen demonstrates anti-tumor efficacy on par with the HIPEC, it is imperative to note that the HIPEC approach, attributable to the DDP and TAX components, may induce pronounced hepatic impairment, potentially exacerbated under hyperthermic conditions. Thus, compared to the traditional HIPEC, the combined strategy of HIPET with AZD1775 offers certain advantages. For patients with hepatic dysfunction or platinum resistance, the combination of HIPET and WEE1i appears to be a clinically valuable option, taking into account both anti-tumor efficacy and safety profiles. We have added this information to the revised manuscript. (Lines 525-535)

Fig S6, related to Fig 8

a Schema of the mouse HIPEC (DDP+TAX) protocol. *b* Representative images (above) and quantification of bioluminescence (below) in C57BL/6 mice after treatment. *c* Quantification and statistical analysis of the abdominal circumference of mice. *d* Tissue sections were stained with Ki67, and γ H2AX. Scale bar, 50 μ m. The quantification of IHC staining scores is shown in the right panel. *e* A plot of ALT, AST, UREA, CREA,

*WBC, RBC, PLT count, and HGB levels in mice following various treatments. Dotted lines indicate the reference range. The p values were determined by a one-way ANOVA (b, c, d), and the results are presented as the mean ± SD (b, c, d) or mean ± SEM (e). ****p < 0.0001, ***p < 0.001, **p < 0.01, *p < 0.05, ns., not significant.*

How long does the hyperthermia + WEE1i last? There is no mention of HIPEC procedure success, did all mice survive to study endpoint?

Response: In vitro, the duration of hyperthermia combined with WEE1 inhibition was 90 minutes (human cells OVCAR8 , A2780, and murine cells ID8) which was consistent with the treatment duration used in clinical HIPEC procedures. Notably, given the unique sensitivity of ID8 cells (murine) to hyperthermia (Figure 3A, S4A-C), we further demonstrated a synergistic anti-tumor effect with 30 minutes of hyperthermia and WEE1i in ID8 cells (Figure S4D-F). This variation of cell sensitivity may stem from differences in species origin and tumor cell genomic features. Numerous studies, including those by Adkins et al. and Morrissey et al., have consistently shown that murine cell lines are more sensitive to hyperthermia compared to human cell lines (PMID: 19845488, PMID: 28638734). To assess the tolerance of tumor-bearing C57 mice to HIPET, we conducted preliminary trials with durations of 30, 60, and 90 minutes. Our data indicated that the mice could effectively withstand 30 minutes of HIPET, with a smooth recovery and minimal post-operative complications within the first 24 hours. However, a 60-minute HIPET hindered their revival, and extending beyond 60 minutes resulted in intraoperative respiratory failure, shock, and, in certain instances, mortality. Given the in vitro sensitivity of the ID8 cell line to hyperthermia and the safety profile of HIPET, we ultimately adopted a 30-minute HIPET treatment in the ID8-C57 model. (Lines 478-482)

After equipment optimization and procedural adjustments, all mice survived to the study endpoint. In our revised manuscript, we have included a description of the HIPEC procedure's success criteria and provide data on the survival of mice, ensuring transparency in reporting the experimental outcomes. (Lines 495-499)

HIPEC patients receive 90 minutes of heated chemotherapy – why was 30 minutes used in this study? need to clarify.

Response: As mentioned, in vitro experiments, the duration of hyperthermia combined with WEE1 inhibition was 90 minutes (human cells OVCAR8 , A2780, and murine cells ID8) which was consistent with the treatment duration used in clinical HIPEC procedures. And, the decision to use a 30-minutes duration of hyperthermia treatment in our animal experiments was based on the limitations of the tolerance of tumor-bearing mice to prolonged heat exposure. We conducted extensive preliminary experiments and optimizations of the perfusion apparatus. The results indicated that C57 tumor-bearing mice could only tolerate 30 minutes of HIPET. Besides, our in vitro experiments demonstrated remarkable synergistic antitumor effects between 30-minute hyperthermia and WEE1 inhibition in ID8 cells (Fig. 8f). So, we applied we ultimately adopted a 30-minute HIPET treatment in the ID8-C57 model in our in vivo experiments. (Lines 478-488)

Upon harvesting tumors, no analysis of tumor tissue was conducted. Why? The IHC analysis is insufficient.

Response: As stated, we utilized a mouse model of ovarian cancer peritoneal dissemination, which is characterized by the presence of numerous small tumor nodules scattered throughout the peritoneal cavity. These nodules are associated with inflammation, adhesions, encapsulation, and fluid accumulation. Due to the nature of this model, it is challenging to isolate individual tumor nodules during the harvesting process, making it impractical to conduct subsequent experiments that require independent tumor lesions. IHC offers a distinctive advantage, allowing researchers to selectively observe molecular staining within tumor regions under the microscope. We also attempted flow cytometry and western blot experiments to analyze the molecular changes post-HIPET and WEE1 inhibition treatments. However, we first encountered a primary challenge: how to precisely isolate tumor tissues without incorporating the adherent normal tissues. While we acknowledge that IHC analysis has its limitations, we believe that the combination of various experimental approaches, including in vitro

studies, functional assays, and histological analyses, provides a comprehensive understanding of the effects of our interventions in the context of the peritoneal dissemination model of ovarian cancer.

There are grammar errors throughout the manuscript, including the abstract. Please get an editor to review.

Response: We appreciate the reviewer's feedback regarding the grammar errors in our manuscript. To address the reviewer's concerns, we engaged a professional editor to enhance the manuscript's language and clarity.

Response to Reviewer #3 (Remarks to the Author)

The study focuses on the effects of combined hyperthermia and WEE1 inhibition in ovarian cancer treatment. The authors present a well-designed study with multiple experiments to assess the efficacy of this combination strategy in ovarian cancer cell lines and mouse models.

Overall, the results of this study are noteworthy and may have a significant impact on the field. The authors aim to demonstrate that the combination of hyperthermia and WEE1 inhibition results in increased DNA damage and cell death compared to either treatment alone in several ovarian cancer cell lines. The methodology is comprehensive, covering multiple cell lines and in vivo models, as well as diverse assays to evaluate the impact of the combination therapy on cell viability, cell cycle, and DNA damage. However, the writing quality throughout the manuscript is suboptimal, with several instances of unclear or confusing language. The manuscript would benefit from thorough proofreading and editing for clarity, grammar, and punctuation.

Response: We are grateful for your thorough evaluation and the insightful suggestions that help us to improve our manuscript. In response to your feedback, we have conducted a thorough proofreading and editing process, ensuring that the writing quality has been substantially improved.

1. Make sure to clearly define all abbreviations when they are first introduced, for example, EdU, γ H2AX, and CCK8.

Response: Thanks so much for your careful review. We have revised the manuscript to explicitly define abbreviations such as EdU, γ H2AX, and CCK8 upon their first mention in the text.

2. Consider reorganizing the paragraph structure to better highlight the key findings and their implications.

Response: We have restructured the paragraphs to provide a clear and concise introduction of the key findings at the end, followed by a detailed explanation of their implications.

3. When introducing the concept of the 'one-two punch' approach, provide a brief explanation of the strategy and its relevance to the study. This will help readers who may not be familiar with the term to better understand the rationale for the drug screening and combination therapy (line 289).

Response: Thank you for your valuable feedback. we have revised the manuscript to include a concise description of the 'one-two punch' approach in the introduction section. Furthermore, we have emphasized the relevance of this approach to our drug screening and combination therapy, highlighting how it can exploit vulnerabilities in cancer cells and maximize treatment efficacy. (Lines 288-301)

Specifically, the 'one-two punch' approach in cancer therapy entails sequentially applying two distinct treatments or interventions to augment the therapeutic outcome. The initial "punch" constitutes a primary treatment targeting tumor cells, whereas the subsequent "punch" serves as a secondary intervention capitalizing on induced vulnerabilities or alterations caused by the initial treatment. Under the conceptual model of 'one-two punch' approach, we further explore whether this hyperthermia induced molecular characteristics would generate a druggable addiction manipulable by a secondary perturbation. As aforementioned, hyperthermia-induced CDK1 hyperactivation, G2/M phase perturbation, replication stress, DNA damage (Fig. 1). So,

we performed candidate drug screening of synergistic potential through combination of HT with 15 well-characterized inhibitors targeting or capitalizing on CDK1 related replication stress or DNA damage pathways individually in OVCAR8, and A2780 cells (Fig. 4A-B). Meanwhile the combination of cisplatin (DDP) and paclitaxel (TAX), which is a standard component of HIPEC, was used as the positive control.

4. The statement "sequential combination of HT and WEE1 inhibitor only occurred before 3, 6, and more than 24 hours in OVCAR8, A2780, and ID8 cells, respectively" could be clearer. It is suggested to rephrase it for better clarity, e.g., "the synergistic lethal effects of sequential combination of HT and WEE1 inhibitor were observed when HT was applied before WEE1 inhibition, with optimal therapeutic windows of 3 hours for OVCAR8 cells, 6 hours for A2780 cells, and more than 24 hours for ID8 cells" (line 380-394).

Response: Thanks so much for your careful review and detailed suggestions provided for the revision of our manuscript. Based on your comments, we have revised the sentence.

5. There are some instances where the language could be improved for better readability. For example, the sentence "Meanwhile, Myt1 overexpression blunted HT-induced DNA damage (reflected by γ H2AX accumulation) and cell death in ID8 and OVCAR8 cells" could be rephrased as "Furthermore, overexpression of Myt1 diminished hyperthermia-induced DNA damage (as indicated by reduced γ H2AX accumulation) and cell death in ID8 and OVCAR8 cells" (line 418-420).

Response: We have rephrased the sentence as suggested. Thank you once again for your thoroughness and dedication in helping us improving the clarity and quality of our work.

6. When presenting the results, consider using a more concise approach. For example, instead of stating "Strikingly, the addition of WEE1i to hyperthermia delivered via intraperitoneal infusion led to profound tumor regression (Fig. 8C-E)," you could say

"Remarkably, combining WEE1i with hyperthermia via intraperitoneal infusion resulted in significant tumor regression (Fig. 8C-E).

Response: Many thanks. We have revised the sentence according to your valuable suggestions. We sincerely appreciate your input and suggestions in teaching us how to enhance the presentation of our research outcomes.

7. Some sentences appear to have typos or errors, such as "...and incubated with and incubated with primary antibody..." and "Cells were subsequently contained with DAPI...". These errors should be corrected to maintain the professional tone of the paper.

Response: We apologize for the oversight and have made the necessary corrections to maintain the professional tone of the paper. Furthermore, we have conducted a comprehensive review of the manuscript to meticulously eliminate any typos and errors.

8. In the HIPEC procedure subsection, the term "emerging" should be replaced with "immersing" when describing warming the perfusate stock bottles.

Response: We have revised it accordingly.

Methodology:

1. Provide more context for the choice of cell lines and justify the selection of the specific hyperthermia conditions (e.g., 42°C for 90 minutes) and WEE1 inhibitor concentrations.

Response: The TP53 gene mutation is prevalent in HGSOC, accounting for approximately 97%. To preliminarily assess whether the activation of CDK1 by hyperthermia and its synergistic anti-tumor effect with WEE1i is dependent on TP53 function, we primarily conducted molecular functional experiments using the TP53 mutant cell line OVCAR8, the TP53 wild-type cell line A2780, and ID8, which lacks the common mutations associated with HGSOC. While our primary molecular studies utilized OVCAR8 and A2780 cell lines, we expanded our validation to 11 ovarian cancer cell lines, including ID8-murine, HOC7, OVCAR8, A2780, OVCAR3,

TOV112D, TOV21G, ES2, CAO3, SKOV3, and OV90-human cells. Representing various ovarian cancer subtypes with diverse mutational backgrounds, these cell lines allowed us to observe that the hyperthermia-induced CDK1 activation and its synergistic effect with WEE1 inhibitor are consistent, irrespective of mutations like ARID1A, ATM, ATR, BRCA1/2, PIK3CA, PTEN, and TP53 (as shown in Revised Fig. S3c). Our findings underscore the universal applicability of this synergistic activity in ovarian cancer, regardless of subtype or genetic background. (Lines 315-317, 321-330, 601-603)

Furthermore, the selected hyperthermia conditions, specifically a temperature of 42°C and a duration of 90 minutes, have been widely utilized in hyperthermia studies and clinical investigations (PMID: 34019431, PMID: 35549912, and PMID: 32377716). These specific parameters have been demonstrated to elicit substantial cellular responses and have shown significant clinical effects. By employing these commonly employed hyperthermia conditions, we aimed to ensure relevance and comparability with previous research findings, as well as establish a foundation for potential clinical translation. (Lines 128-129)

Additionally, the WEE1 inhibitor concentrations were selected based on our published studies (PMID: 31185210, PMID: 35619328) and preliminary dose-response experiments conducted in our laboratory. Overall, the concentration of AZD1775 (Adavosertib) we selected is the minimum concentration that can activate CDK1 in various cell lines. Assessing the sensitizing effect of hyperthermia on the drug at the lowest AZD1775 concentration helps to avoid a potential bias where a strong monotherapeutic effect of AZD1775 might overshadow the combined effect. Fang et al.'s study demonstrated that in the OVCAR8 cell line, the minimum concentration of AZD1775 required to significantly reduce cdc2Y15 (CDK1-pY15) was 240 nM (PMID: 31185210). Conversely, Xiao et al. identified that both A2780 and ID8 cell lines required an 800nM concentration of AZD1775 to achieve a similar downregulation (PMID: 35619328). Hence, in our study, OVCAR8, A2780, and ID8 cells were administered with AZD1775 at concentrations of 250nM, 800nM, and 800nM, respectively. Specifically, in the section of the manuscript where the combination

therapy of hyperthermia and AZD1775 is first introduced, we have incorporated the rationale behind the dosage selection and provided references to pertinent literature.

(Lines 307-311)

2. Describe in more detail the experimental procedures, including the number of replicates, control conditions, and any potential confounding factors considered.

Response: In the revised manuscript, we have provided a more detailed description of the experimental procedures, including the number of replicates performed for each experiment. Besides, we have presented the repeated group data using dot plots, allowing for an intuitive visualization of the number of repetitions in each group (revised Figs). We have also elaborated on the control conditions implemented to ensure accurate comparisons and minimize potential confounding factors

3. Include details about the antibodies used for immunoblotting and immunohistochemistry, such as dilution ratios, manufacturers, and catalog numbers.

Response: We have provided information on the antibody's dilution ratios, manufacturers, and catalog numbers used in the "Supplementary Table 5" subsection under the "Methods" section.

4. Justify the use of the specific mouse model and discuss any potential limitations or caveats associated with this model.

Response: In the revised manuscript, we've provided a detailed rationale for our choice of mouse model and discussed its potential limitations. (Lines 607-631)

5. In the IHC staining subsection, the authors should specify the duration of blocking in 5% BSA with 0.1% Triton X-100 in PBS.

Response: The samples were blocked for 1 hours to ensure effective blocking of non-specific binding sites and enhance the specificity of antibody binding.

6. In the establishment of intraperitoneal metastatic ovarian tumors in mice subsection, the authors should provide the number of mice used for this part of the study.

Response: We utilized a total of 23 mice for the intraperitoneal tumor model. Two weeks post ID8 cell injection, three mice were assessed with IVIS and laparotomy to confirm tumor growth. The remaining 20 mice were then randomly divided into four groups for intraperitoneal perfusion and subsequent drug treatments. In our revised manuscript, we have provided data on the survival of mice, ensuring transparency in reporting the experimental outcomes. (Lines 495-499, Experimental design in vivo subsection from Methods)

7. Provide clearer explanations for some parts of the methods, such as the rationale behind the choice of AZD1775 dosages and administration schedules for the in-vivo experiments.

Response: In the revised manuscript, we have provided clearer explanations for the rationale behind the choice of AZD1775 dosages and administration schedules for the in-vivo experiments. (Experimental design in vivo subsection from Methods)

8. In the experimental design in vivo subsection, it is unclear how many days after drug administration the fluorescence images were conducted. The authors should specify this information.

Response: We performed IVIS imaging to assess the tumor burden in the mice at the experimental endpoint, 21 days after the administration. (Lines 495-499)

9. In the statistical analysis subsection, the authors should provide information on the normality of the data distribution and the assumptions met for the chosen statistical tests.

Response: Thank you for your valuable feedback. We have added the relevant information to the Statistical analysis section.

Results and data presentation

1. When presenting the results for Figure 1, clarify the mechanism by which hyperthermia leads to increased DNA damage and how this is related to WEE1 inhibition. Please also clarify the relationship between the endogenous DNA damage response, cell cycle phase and the response to exogenous DNA damage.

Response: We have provided a more detailed explanation of the mechanism by which hyperthermia leads to increased DNA damage and how it is related to WEE1 inhibition. (Lines 211-214) The endogenous DNA damage response centers on intracellular genomic maintenance and repair mechanisms, such as base excision repair and chromatin remodeling (PMID: 28698521, PMID: 33540225, PMID: 12760027). In contrast, the exogenous response focuses on detecting DNA damage from outside factors and involves sensing mechanisms (ATM, ATR, DNA-PK), cell cycle regulation, and apoptosis (PMID: 25423595). Together, these pathways ensure genomic stability and address both endogenous and exogenous DNA damages. Hyperthermia, through CDK1 activation, induces replication stress, DNA damage, and cell cycle dysregulation, affecting exogenous DNA repair pathways. This effect is further amplified when combined with WEE1 inhibitors, leading to enhanced DNA damage and increased cytotoxicity. This synergy mainly targets exogenous DNA repair pathways, independent of the endogenous DNA repair mechanism's functionality. We have added the relevant information to the figure 1 section. (Lines 595-598)

2. When presenting the results of the drug screening, consider providing more information on the specific inhibitors tested and their targets. This will give readers a better understanding of the scope of the screening and the potential mechanisms of action of the inhibitors.

Response: In Fig4A, we presented the targets of each inhibitor and integrated them into a comprehensive diagram of CDK1-related DNA damage repair and cell cycle pathways. Following the reviewer's feedback, we recognized the potential insufficiency of this representation. Therefore, in the Methods section, we've included **Supplementary Table 2** to display the IC50 of each inhibitor from cell-free assays, alongside target specificity information and potential secondary targets.

3. It would be helpful to specify the cell lines used in the experiments for this section, as this information is not provided (line 349 -368).

Response: We have provided specific details about the cell lines used in the section.

4. The authors should provide a brief rationale for exploring the different combination strategies (sequential or concurrent). This would help readers understand the significance of these experiments.

Response: In clinical settings, there are various ways to administer combination therapy, including concurrent administration and sequential administration. The 'one-two punch' approach in cancer therapy entails sequentially applying two distinct treatments or interventions to augment the therapeutic outcome. The initial "punch" constitutes a primary treatment targeting tumor cells, whereas the subsequent "punch" serves as a secondary intervention capitalizing on induced vulnerabilities or alterations caused by the initial treatment. To implement the 'one-two punch' approach effectively, it is pivotal to first delineate the sequence of the two interventions - hyperthermia or WEE1 inhibition. Additionally, once the sequence is ascertained, the timing between the primary and secondary interventions must be optimized. This ensures that the second intervention is administered promptly after the maximum disruption caused by the first, maximizing the synergistic anti-tumor effects of the combined therapy. The relevant explanations have been included in the revised manuscript. (Lines 392-398)

Discussion and conclusions:

1. Explore potential off-target effects of the WEE1 inhibitor or other factors that could influence the response to hyperthermia and discuss how these may have impacted the study outcomes.

Response: We utilized AZD1775, a highly selective WEE1 inhibitor (PMID: 32958072, PMID: 32688199). Additionally, to rule out potential off-target effects of AZD1775, we carried out siRNA-mediated WEE1 knockdown in OVCAR8. The results suggest that the combined effect of hyperthermia and WEE1 inhibition is not due to off-target

effects of AZD1775 (as shown in the newly added Fig. S3d, e). Indeed, we will also consider other factors that could potentially influence the response to hyperthermia, such as the cellular microenvironment, tumor heterogeneity, and inter-individual variability. By discussing these potential confounding factors, we aim to provide a comprehensive analysis of the study outcomes and their interpretation, thereby enhancing the scientific rigor and validity of our findings. (Lines 317-320)

Fig S3, related to Fig 4

d Western blot analysis of WEE1 expression in OVCAR8 cells transfected with scramble siRNA (NC) or WEE1 siRNA. *e* Cell viability was determined by CCK8 assay in OVCAR8 cells treated as indicated. The *p* values were determined by a one-way ANOVA, and the results are presented as the mean \pm SD. *****p* < 0.0001, **p* < 0.05.

2. Elaborate on potential limitations of the study, such as the use of cell lines and mouse models, which may not fully represent the complexity of human ovarian cancer.

Response: We have provided a comprehensive discussion on the limitations associated with the use of cell lines and mouse models, emphasizing the need for further studies using more clinically relevant models. Specifically, our study highlights the potent synergistic anti-tumor activity of combining hyperthermia with WEE1 inhibition. This combined effect has been robustly validated in vitro, spanning diverse ovarian cancer cell lines and clinical primary samples. In our study, we used the ID8 cell line for a mouse ovarian cancer peritoneal dissemination model, a common choice in preclinical research. However, it's crucial to note that the ID8 model might not fully reflect the behavior and characteristics of human ovarian cancer, including differences in growth rate, aggressiveness, metastasis, and treatment response. We attempted to apply OVCAR8-based ovarian cancer model in Nude mice but faced challenges due

to their lower heat tolerance compared to C57 mice, resulting in complications from a hyperthermic intraperitoneal perfusion. Recognizing this limitation, we're now exploring safer hyperthermal methods and considering patient-derived xenograft (PDX) models to better represent human ovarian cancer complexity. This limitation and ongoing efforts are discussed in our manuscript. (Lines 625-635)

3. Discuss the clinical relevance of the findings and any potential challenges or considerations for translating the results into clinical practice.

Response: Thank you for your suggestion. Historically, clinical trials of hyperthermic intraperitoneal chemotherapy (HIPEC) have yielded suboptimal benefits, potentially due to an insufficient understanding of the tumor cell changes induced by hyperthermia, leading to a lack of diverse combined therapeutic strategies. Our research aims to provide new insights for hyperthermic intraperitoneal therapy (HIPET), with a particular focus on the synergistic effects between hyperthermia and WEE1 inhibition, intending to identify potential targets that can enhance therapeutic outcomes, and explore the potential of combining HIPET with precision-targeted therapeutic drugs, aspiring to transition from the traditional chemotherapy-associated HIPET towards an era of targeted precision treatment. The clinical relevance of the findings in this study is significant in the context of improving the treatment of ovarian cancer. However, several challenges and considerations need to be acknowledged when translating these results into clinical practice.

Dosage and Timing: While our study offers insights into the effectiveness of the combined treatment in preclinical models, determining the optimal dosage and timing for human subjects requires further investigation.

Potential Side Effects: The potential for off-target effects and the systemic implications of combining hyperthermia with WEE1 inhibition need to be carefully assessed in clinical trials to ensure patient safety.

Patient Selection: The efficacy of the combined treatment may vary based on individual patient genetics, tumor type, and stage. Thus, patient stratification and identifying potential biomarkers for responsiveness would be crucial.

Equipment and Training: The application of hyperthermia in a clinical setting requires specialized equipment and trained personnel to ensure that the treatment is delivered safely and effectively.

Economic Considerations: The cost implications of integrating hyperthermia into standard treatment regimens, especially in resource-limited settings, should be considered.

We have now incorporated a section in the discussion of our revised manuscript that elaborates on the clinical relevance of our findings and addresses the potential challenges and considerations for translation into clinical practice. (Lines 631-635)

In summary, the manuscript presents a valuable contribution to the field of ovarian cancer research, with compelling evidence supporting the potential of combining hyperthermia and WEE1 inhibition as a therapeutic strategy. However, the manuscript requires significant revisions to address the concerns mentioned above, particularly in terms of writing quality and clarity. I recommend that the authors revise and resubmit their manuscript to address these concerns before it can be considered for publication.

Response: Thank you once again for your valuable feedback. We have made concerted efforts to address each of the specific concerns you raised.

REVIEWERS' COMMENTS

Reviewer #2 (Remarks to the Author):

Yang et al., have revised their manuscript titled "Comprehensive multi-omics analysis revealed WEE1 as a synergistic lethal target with hyperthermia through CDK1 super-activation" focused on identifying the mechanisms of action of hyperthermic intraperitoneal chemotherapy (HIPEC) in epithelial ovarian cancer. The studies are greatly improved with more focus on more limited to the direct interpretation of the studies. The limitations of obtaining clinical specimens are understood and should be considered in future analyses.

Reviewer #3 (Remarks to the Author):

Dear authors,

I am writing in reference to the revised manuscript titled "Comprehensive multi-omics analysis revealed WEE1 as a synergistic lethal target with hyperthermia through CDK1 super-activation" submitted to Nature Communications.

Upon reviewing the revised submission, I note the following:

1. The authors have thoroughly addressed the initial review concerns.
2. The research, emphasizing the synergistic effects of hyperthermia and WEE1 inhibition in ovarian cancer therapy, is methodically conducted and holds clinical significance.

Given the revisions and the potential importance of the findings, I recommend considering the manuscript for publication. However, this is contingent upon the concurrence of other reviewers. Their comprehensive evaluation will be paramount for the final decision.